# Developmental cues are encoded by the combinatorial phosphorylation of *Arabidopsis* RETINOBLASTOMA-RELATED protein RBR1

Jorge Zamora-Zaragoza [ID] [1,2], Katinka Klap[1], Jaheli Sánchez-Pérez [ID] [3], Jean-Philippe Vielle-Calzada [ID] [3], Viola Willemsen [ID] [1] & Ben Scheres [ID] [1,2 ✉]

## Abstract

**RETINOBLASTOMA-RELATED (RBR) proteins orchestrate cell division, differentiation, and survival in response to environmental and developmental cues through protein–protein interactions that are governed by multisite phosphorylation. Here we explore, using a large collection of transgenic RBR phosphovariants to complement protein function in *Arabidopsis thaliana*, whether differences in the number and position of RBR phosphorylation events cause a diversification of the protein's function. While the number of point mutations influence phenotypic strength, phosphosites contribute differentially to distinct phenotypes. RBR pocket domain mutations associate primarily with cell proliferation, while mutations in the C-region are linked to stem cell maintenance. Both phospho-mimetic and a phospho-defective variants promote cell death, suggesting that distinct mechanisms can lead to similar cell fates. We observed combinatorial effects between phosphorylated T406 and phosphosites in different protein domains, suggesting that specific, additive, and combinatorial phosphorylation events fine-tune RBR function. Suppression of dominant phospho-defective RBR phenotypes with a mutation that inhibits RBR interacting with LXCXE motifs, and an exhaustive protein–protein interaction assay, not only revealed the importance of DREAM complex members in phosphorylation-regulated RBR function but also pointed to phosphorylation-independent RBR roles in environmental responses. Thus, combinatorial phosphorylation defined and separated developmental, but not environmental, functions of RBR.**

**Keywords** RETINOBLASTOMA-RELATED; RBR Phosphorylation; Cell Division; Cell Death; LXCXE Motif
**Subject Categories** Development; Plant Biology

## Introduction

Multicellular organisms coordinate cell division and differentiation in space and time to ensure proper development (Gutierrez, 2005; Sablowski and Carnier Dornelas, 2014). When the environment is variable, external cues need to be incorporated in this coordination process. Orchestration of developmental programs and environmental responses becomes particularly challenging in sessile species like plants. The multifunctional protein RETINOBLASTOMA-RELATED1 (RBR) of Arabidopsis, a homolog of the human Retinoblastoma (RB) susceptibility gene product (pRb), acts as an integrator of environmental cues and internal programs into cell fate decisions (Gutierrez, 2005; Harashima and Sugimoto, 2016).

RBR belongs to the pocket-protein family, which function as protein interaction platforms that bring together multiple transcriptional and chromatin regulators, thus controlling genetic programs (Dick and Rubin, 2013; Gutzat et al, 2012). For example, RBR controls cell division by interacting with and inhibiting activation of the S-phase program by E2F-DP heterodimeric transcription factors. Stable repression of cell cycle genes leads to a quiescent state achieved by the DREAM complex (named after its constituents D̲P-E̲2F-R̲BR and the Multivuvla B complex, M̲uvB), which regulates chromatin structure and DNA methylation (Kobayashi et al, 2015; Ning et al, 2020). RBR-mediated repression is alleviated by phosphorylation, primarily by CYCLIN-DEPENDENT KINASES (CDK). CDKA associates with D-type CYCLINS (CYCD) which target CDKA-CYCD phosphorylation activity to RBR through the CYCD LXCXE motif, thereby releasing E2F repression. RBR also controls formative divisions through similar mechanisms (Cruz-Ramírez et al, 2012; Han et al, 2018; Matos et al, 2014; Weimer et al, 2018) to couple cell division and fate decisions.

The involvement of RBR, and distinct CYC-CDKs and CDK inhibitors (CKI) in both developmental and stress-related processes (Biedermann et al, 2017; Gutierrez, 2005; Horvath et al, 2017; Perilli et al, 2013; Sablowski and Carnier Dornelas, 2014; Wang et al, 2014a; Weimer et al, 2016; Wen et al, 2013; Yi et al, 2014; Zhao et al, 2017), some of which occur simultaneously, points to a central, as yet unspecified role for RBR phosphorylation in the integration of signaling inputs to orchestrate coordinated cell behavior.

[1]Laboratory of Cell and Developmental Biology, Department of Plant Sciences, Wageningen University and Research, 6708 PB Wageningen, The Netherlands. [2]Rijk Zwaan Breeding B.V., Department of Biotechnology, Eerste Kruisweg 9, 4793 RS Fijnaart, The Netherlands. [3]Laboratorio Nacional de Genómica para la Biodiversidad, Centro de Investigación y de Estudios Avanzados del Instituto Politécnico Nacional, 36824 Irapuato, Guanajuato, Mexico. ✉E-mail: ben.scheres@wur.nl

Both human pRb and Arabidopsis RBR contain 16 putative CDK phosphorylation sites, mostly located in the inter-domain regions. Crystal structures of pRb fragments demonstrate that specific phosphorylated residues induce discrete structural changes that promote different intramolecular interactions to either prevent or compete with intermolecular interactions (Burke et al, 2010, 2012). Biochemical characterization of the effect of specific phosphorylation residues on the interaction with E2Fs and with the LXCXE motif indicates that the phosphosites contribute differentially to regulate pRb-protein interactions (Burke et al, 2010, 2014; Rubin et al, 2005). These observations led researchers to speculate that a 'phosphorylation-code' exists, whereby distinct phosphorylation events generate unique structural changes to influence pRb binding properties and functions (Munro et al, 2012; Rubin, 2013). Although attractive, the phosphorylation code hypothesis requires experimental evidence, particularly in plants.

Here, we took a systematic approach to study the biological relevance of RBR phosphorylation. Using a large collection of transgenic loss- and gain of function point mutations in RBR phosphosites, we set out to disentangle RBR roles by specific phosphorylation combinations. We found that, whereas phosphorylation within the N-domain of RBR gives less prominent effects in general, phosphorylation within the pocket domain has a greater influence on meristem cell proliferation, and the C-terminal region markedly associates with the stem cell maintenance activity of RBR. Surprisingly, specific combinations of phospho-defective mutations can lead to hyper-active variants of RBR that promote cell death while restraining proliferation; and the contribution of a phosphosite to the function of RBR varies according to the phosphorylation state of other sites. Finally, we show strong dominant effects of non-phosphorylatable RBR variants and that these can be suppressed by interfering with their ability to bind LXCXE motif-containing proteins like the DREAM complex members of the TCX5/6/7 clade. Our findings provide new insights on the conserved mechanisms underlying RBR function, uncovering the combinatorial nature of RBR phosphorylation-dependent control of cell division, differentiation and survival, while pointing to a phosphorylation-independent role in stress and environmental responses.

## Results

### A system to study phosphovariants by circumventing early lethality

The substantial knowledge on plant RETINOBLASTOMA-RELATED (RBR) proteins derives from expression studies, null or hypomorphic alleles, and up- or downregulation of the gene (Ach et al, 1997; Borghi et al, 2010; Chen et al, 2011; Cruz-Ramírez et al, 2013; Ebel et al, 2004; Grafi et al, 1996; Gutzat et al, 2011; Perilli et al, 2013; Wachsman et al, 2011; Wildwater et al, 2005; Xie et al, 1996). However, the functional outcome of RBR phosphorylation remains largely unexplored, in spite of being assumed to be a major regulatory mechanism of RBR activity. We approached this subject by constructing a representative collection of transgenic RBR phosphovariants comprising all putative CDK phosphorylation sites (Desvoyes and Gutierrez, 2020; Desvoyes et al, 2014) (Fig. 1).

Tests of all possible phosphorylation states on 16 sites would entail the construction of $3^{16}$ (~43 million) variants, so we simplified the analysis by taking a domain approach. Briefly, the coding sequence of RBR was split into three combinable modules named as "N", "P", and "C" (after the N-terminal, AB-Pocket, and C-terminal protein domains), each bearing a subset of phosphosites in the one of three states: phosphorylatable (wild-type), phospho-defective, and phospho-mimetic, the later resembling constitutive phosphorylation (Antonucci et al, 2014; Chen et al, 2017; Dissmeyer and Schnittger, 2011; Sanidas et al, 2019; Wang et al, 2014b). These are depicted by "0", "-" and "+" signs, respectively (or by a colored circles code in figures, Figs. 1A,B and EV1A). We refer to each RBR phospho-variant as the specific combination of modules, with a superscript indicating the total number of mutated sites. For example, [N0,P0,C0]$^0$ refers to the fully phosphorylatable RBR, while [N0,P-,C0]$^5$ and [N0,P+,C0]$^5$ respectively denote phospho-defective and phospho-mimetic versions of the five phosphosites in central module only (Fig. 1B). We refrained from combining phospho-defective with phospho-mimetic modules and assembled all other possible variants with the native RBR promoter and a SCFP3A C-terminal tag to select comparable expression levels of transgenic plants based on the SCFP3A intensity (Figs. 1B,C and EV2B). All RBR phosphovariants were transformed into plants homozygous for the amiGO-RBR genetic construct (hereafter, amiGO; Fig. 1C,D), an artificial microRNA driven by the 35S promoter that selectively down-regulates endogenous RBR transcripts only after the gametophyte stage and completion of early embryogenesis (Cruz-Ramírez et al, 2013). This late reduction in RBR levels bypasses its requirement in early developmental stages and enhances both cell proliferation and death similar to a true null *rbr* clone (Wachsman et al, 2011). Most but not all RBR phosphorylation variants accumulate at similar levels, which was assessed both by the SCFP3A intensity and western-blotting the transgenic product (Fig. EV2A). However, the lower accumulation of some variants, which might reflect counter-selection by fertility issues (shown later), does not preclude their strong phenotypes, suggesting that the effect of phospho-site mutations on RBR activity is not solely due to differences in protein accumulation. Through analysis of the complementation capacity of all viable homozygous transgenic variants at 6 days after sowing (das), when the amiGO phenotypes were fully penetrant (Fig. EV2B), we could assess the effect of site-specific mutant combinations in RBR.

### C-region phosphorylation inhibits RBR-mediated restriction of stem cell (SC) division

Since the downregulation of RBR leads to supernumerary QC and SC divisions (Cruz-Ramírez et al, 2013; Wildwater et al, 2005), we first asked whether stem cell niche (SCN) proliferation is affected by specific RBR phosphorylation events. Aberrant division planes hinder lineage identification in the absence of markers, so we quantified the pooled number of QC, cortex and endodermis initials (CEI), and columella stem cells (CSC) to explore the effect of the phosphovariants in SCN maintenance.

All but two unviable phospho-defective variants (indicated as ⊠ in Fig. 2A) complemented the SCN overproliferation phenotype induced by the amiGO at a level at least equal to the complementation by wild-type RBR (Fig. 2A,B), consistent with dephosphorylated RBR acting as the repressor of SCN activity. Among these variants, the [N-,P-,C0]$^{12}$

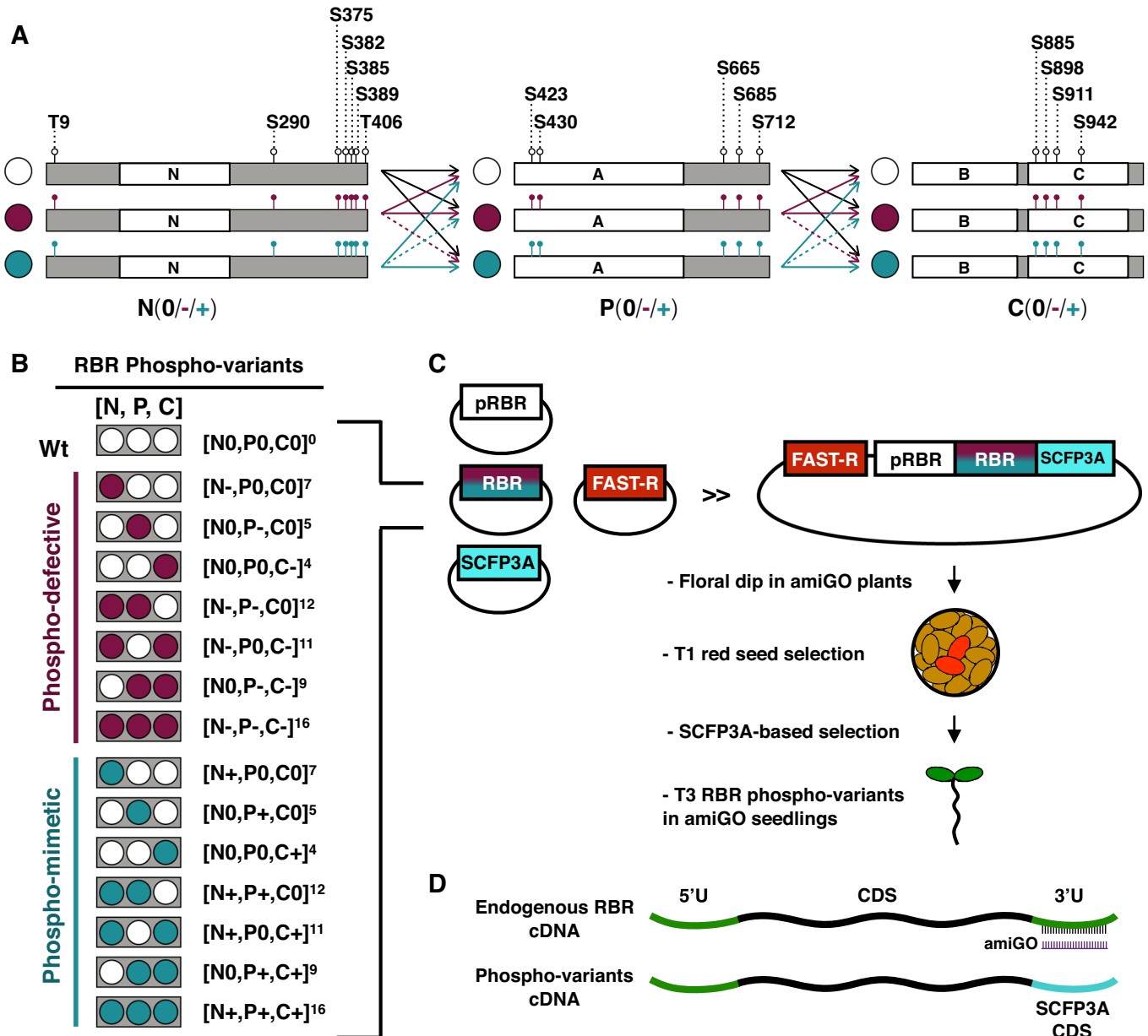

**Figure 1. A system to study phosphovariants by circumventing early lethality.**

(A) We cloned three combinable fragments (N, P, and C) comprising the full-length RBR cDNA into the level -1 vector of the Golden Gate MoClo toolbox (Engler et al, 2014). White boxes represent RBR protein domains (N-terminal, A- and B-**P**ocket subdomains, and C-terminal) as predicted by the pfam server (https://pfam.xfam.org/), and gray boxes represent unstructured protein regions. Note that the P module contains all phosphosites within the Pocket domain, but due to cloning convenience, the C-module encodes the B-pocket sub-domain. Each level -1 module encodes a subset of phosphorylation sites indicated by empty or colored lollipops, and the corresponding the amino acid residue (S/T) position. Since all phosphosites within a module are in the same state, we use colored circles and the signs "0", "-", and "+" to denote the module state: white/0 for phosphorylatable (Ser/Thr), dark red/- for phospho-defective (Ala), and teal/+ for phospho-mimetic (Asp or Glu). All codon changes are listed in Appendix Table S2. All combinations are possible (arrows), but we avoided phospho-defective with phospho-mimetic combinations (dotted arrows). (B) Color code and text nomenclature of phosphovariants. (C) Generation and selection of transgenic RBR phosphovariants plants. Modules were assembled into the level 0 vector to generate full-length RBR phosphovariants CDS, subsequently assembled with RBR promotor, the CDS of SCFP3A fluorescent tag, and a terminator (not illustrated) into Level 1 constructs. Level 2 constructs containing the FAST-R selection cassette and the corresponding phospho-variant were transformed in 35::amiGO-RBR plants (amiGO). T1 seedlings pre-selected by the red seed coat were selected for the best SCFP3A intensity and taken to T3 generation for complementation analysis. (D) amiGO selectively down-regulates endogenous RBR transcripts by targeting the 3'-UTR, which is absent in the transgenic RBR:SCFP3A CDS. See Fig. EV1 for the full list of modules and phosphovariants generated.

# A

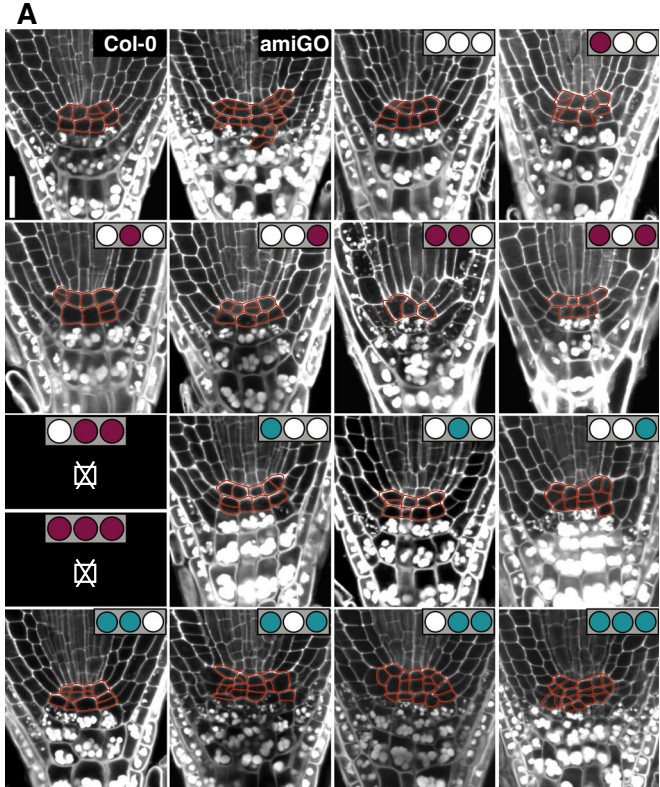

# B

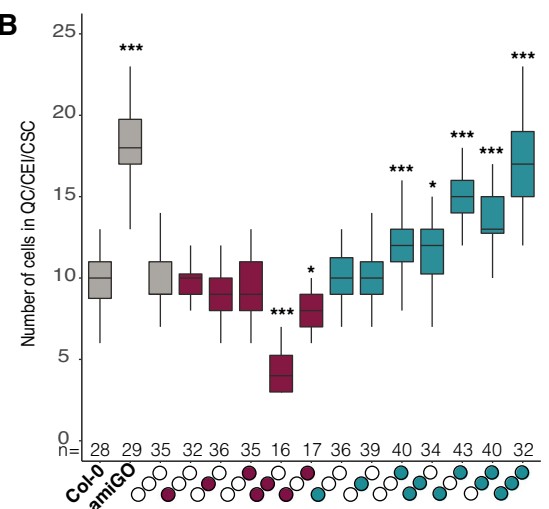

**Figure 2. C-region phosphorylation inhibits RBR-mediated restriction of stem cell (SC) division.**

(A) Representative confocal images of modified pseudo-Schiff propidium iodide (mPS-PI) stained root tips of the indicated genotypes showing starch granules accumulation in differentiated cells. Sub-panels marked with ⊠ correspond to lethal genotypes. Scale bar, 20 μm. (B) Box plot of pooled QC cells, CEI, and CSC number excluding cells with evident starch granules accumulation. Data from two biological replicates. In this and all following box plots, the lower and upper bounds of the box are at the first and third quartiles, respectively, the center line at the median, and the minima and maxima values are indicated by whiskers. n denotes total number of scored roots. Dunett's test against Col-0, ***$P < 0.001$, *$P < 0.05$.

combination of phospho-defective residues in N and P domains, which had no effect on their own, showed a higher level of repression indicating that phosphorylation in these domains have a combinatorial and possibly additive effect. Thus dephosphorylation is essential for the role of RBR in normal maintenance of the SCN, and single-domain dephosphorylation is insufficient for maximal RBR-mediated repression.

Consistent with a role for dephosphorylated RBR in repressing SCN activity, several variants with a phospho-mimetic module failed to suppress SCN overproliferation. Overproliferation never exceeded that seen in the amiGO background, but increased with the number of domains containing phospho-mimetic residues. Thus, phosphorylation in more than one RBR domain is needed to relieve the repression of SCN divisions. However, $[N0,P0,C+]^4$ revealed incomplete repression of SCN activity similar to $[N+,P+,C0]^{12}$, despite having fewer phospho-mimetic residues, indicating that phosphorylation is not simply additive and that sites in the C domain have a greater influence on SCN regulation than those in the N and P domains. Taken together, a range of phospho-defective and phospho-mimetic mutant combinations reveals additive but differential contributions to the regulation of SCN activity by RBR phosphosites in all three protein domains.

## Meristem size maintenance depends most strongly on Pocket domain phosphorylation

Similar to their effect on SCN activity, down- and upregulation of RBR have opposite effects on root meristem size (Perilli et al, 2013), reflecting control of cell division. To elucidate whether cell division activity also depends on specific RBR phosphorylations in the root meristem, we measured the effects of the phospho-site variants on the size of the transit-amplifying cell pool in the meristem. As expected, amiGO meristems were slightly longer and contained more cells than Col-0, which could be fully restored using the Wt RBR version $[N0,P0,C0]^0$ (Fig. 3A,B).

With the exception of $[N0,P0,C+]^4$, which contained more cells but with a meristem length not significantly different than the Wt, phospho-mimetic variants were all able to complement the amiGO meristem phenotype, indicating that almost any additional activity of RBR mitigates the slight meristem size increase observed in the amiGO root meristem. Unlike observed for the SCN proliferation phenotype, no other variant containing the C+ module exhibited significant changes (Fig. 3A,B). Conversely all phospho-defective variants reduced amiGO-induced overproliferation in the meristem (Fig. 3B). However, in this case, unlike the single-domain N- or C-phospho-defective variants, the phospho-defective Pocket domain alone in $[N0,P-,C0]^5$ was sufficient to over-complement the amiGO mutants, exhibiting shorter meristems than the Wt, without significant effect on the SCN (Figs. 2B and 3B,C). Our data indicate distinct effects for RBR phosphorylation sites in control over SCN and meristem proliferation, with a larger role for the C-region phosphorylation in the SCN, and for the Pocket domain in the transit amplifying cells of the meristem.

Interestingly, $[N0,P0,C+]^4$ and $[N-,P0,C-]^{11}$ did not affect meristem length despite presenting more cells, respectively (Fig. 3B). Similar observations have been reported for mutants or overexpressors of genes that influence RBR phosphorylation, such as CDKA, CYCD3, and CDK inhibitors (CKIs), where cell size and cell division compensate each other to maintain meristem and/or

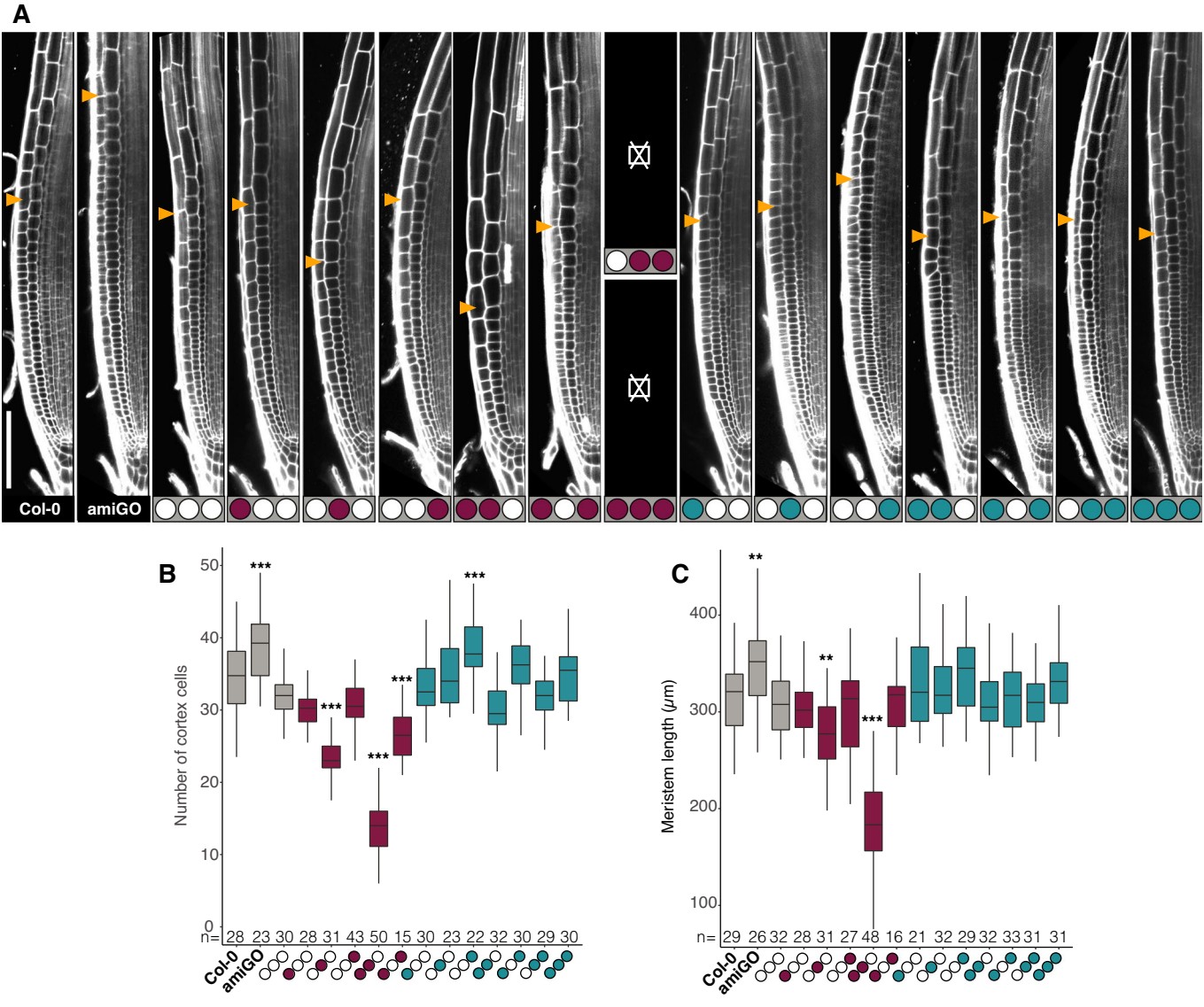

**Figure 3. Meristem size maintenance depends most strongly on Pocket domain phosphorylation.**

(A) Representative confocal images of root meristems of the indicated genotypes; yellow arrowheads mark the end of the meristem proliferation zone. Sub-panels marked with ⊠ correspond to lethal genotypes. Scale bar, 100 μm. (B, C) Box plots of meristem proliferation and size quantified as the number of cortex cells (B) and length (C) from QC to the first rapidly elongating cortex cell. Data from two biological replicates presented as median (center line), interquartile range (box) and minima and maxima values (whiskers), *n* denotes total number of scored roots. Dunett's test against Col-0, ***P < 0.001, **P < 0.01.

organ size (Cheng et al, 2015; Horiguchi and Tsukaya, 2011; Truskina and Vernoux, 2018). But compensatory effects are limited, as the presence of the phospho-defective P module always resulted in a reduced meristem length, similar to drastic changes in CKI expression (Cheng et al, 2015; Noir et al, 2015; Verkest et al, 2005). Thus, phosphorylation of the Pocket domain is particularly important to maintain meristem size.

## Suppression of cell death is rescued by all but two distinct RBR phosphovariants

Spontaneous cell death in the root tip constitutes a hallmark of reduced RBR activity (Cruz-Ramírez et al, 2013; Wildwater et al,

2005), likely due to the inability to cope with intrinsic DNA damage (Biedermann et al, 2017; Horvath et al, 2017). We examined the protective role of RBR phosphorylation variants using propidium iodide staining (PI), which permeates only dead cells visualized as red spots. As expected, all amiGO roots presented dead cells, while Col-0 and the vast majority of the phosphovariants had around 25% or less root tips with dead cells. Two phosphovariants reached a comparable cell death frequency to amiGO seedlings (Fig. 4A,B). The full phospho-mimetic variant $[N+,P+,C+]^{16}$ fits the paradigm of hyper-phosphorylated RBR being inactive. In agreement with this, $[N+,P+,C+]^{16}$ also presented overproliferation of SCN and meristematic cells (Figs. 2 and 3A,B), supporting the supposition that phospho-mimetic mutations inactivate RBR.

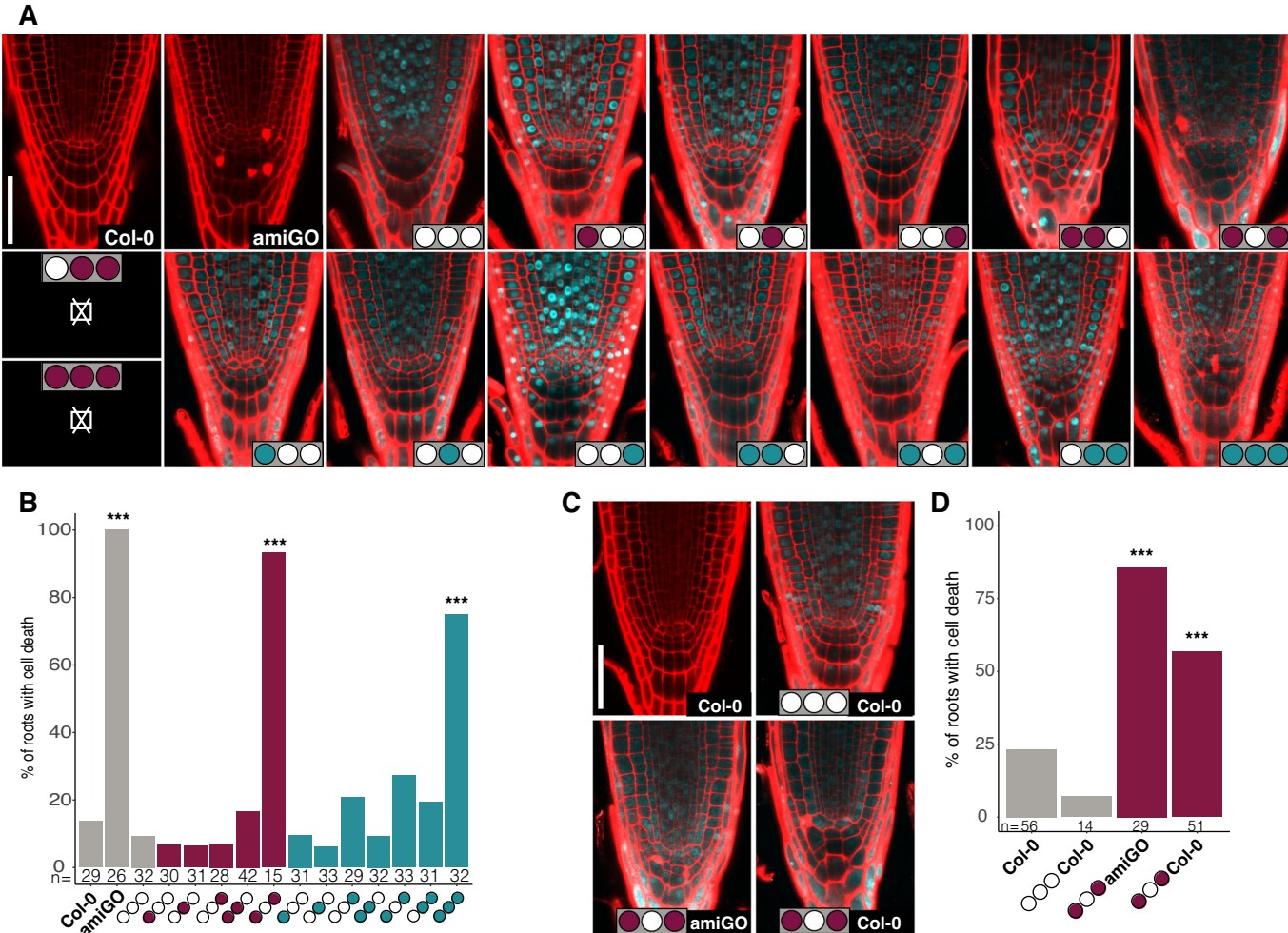

**Figure 4. Suppression of cell death is rescued by all but two distinct RBR phosphovariants.**

(A) Representative confocal images of PI-stained (red) root tips of the indicated SCFP3A-tagged (cyan) phosphovariants. Sub-panels marked with ⊠ correspond to lethal genotypes. (B) Bar graphs from (A) showing percentage of root tips with dead cells. (C) Representative confocal images similar to (A) (for red and cyan colors) of the indicated genotypes with the genetic background indicated in black text boxes. In (A, C) red spots in the SCN area correspond to cell death, as PI selectively stains dead cells; scale bars, 50 μm. (D) Bar graphs from (C) showing percentage of root tips with dead cells. In (B, D), data from two biological replicates presented as means, n denotes total number of scored roots. Fisher's test against Col-0, ***$P < 0.001$.

However, the phospho-defective variant [N-,P0,C-][11] presented a striking outcome for a RBR isoform presumed to be active although not able to be phosphorylated in N nor C-terminal domains. [N-,P0,C-][11] over-complemented the amiGO cell proliferation phenotypes (Figs. 2 and 3A,B) but failed to promote cell survival, in contrast with [N0,P-,C0][5] and [N-,P-,C0][12], that also over-complemented cell proliferation but fully restored the cell death phenotype (Figs. 2B and 4A,B). In addition, some phospho-mimetics that failed to restrain SCN proliferation still suppressed cell death. Thus, cell proliferation is always promoted by RBR phosphorylation to a greater or lesser extent according to specific phosphosites, but cell death emerges either upon constitutive RBR hyper-phosphorylation or with a specific combination of un-phosphorylated sites, implying two different mechanisms for RBR-promoted cell survival.

Since phospho-defective [N-,P0,C-][11] efficiently restrains cell division, we asked whether the cell death phenotype is caused by the activity of RBR, or results from an impaired protective function. We out-crossed the amiGO background to assess the effect of [N-,P0,C-][11] and [N0,P0,C0][0] in the presence of endogenous RBR. While four copies (endogenous and transgenic) of wild-type RBR conferred a protective effect, more than 50% of the [N-,P0,C-][11] roots still displayed dead cells in the Col-0 genetic background (Fig. 4C,D). However, in the amiGO background the frequency increased to more than 80% (Fig. 4C,D), indicating that [N-,P0,C-][11] is an RBR active isoform triggering the cell death program, possibly counteracted by the endogenous RBR.

## Combining full domain with single-site phospho-site indicates that combinatorial phosphorylation regulates RBR

The distinct contributions of phosphosites in different RBR protein domains to cell division phenotypes (C-region sites to SNC activity

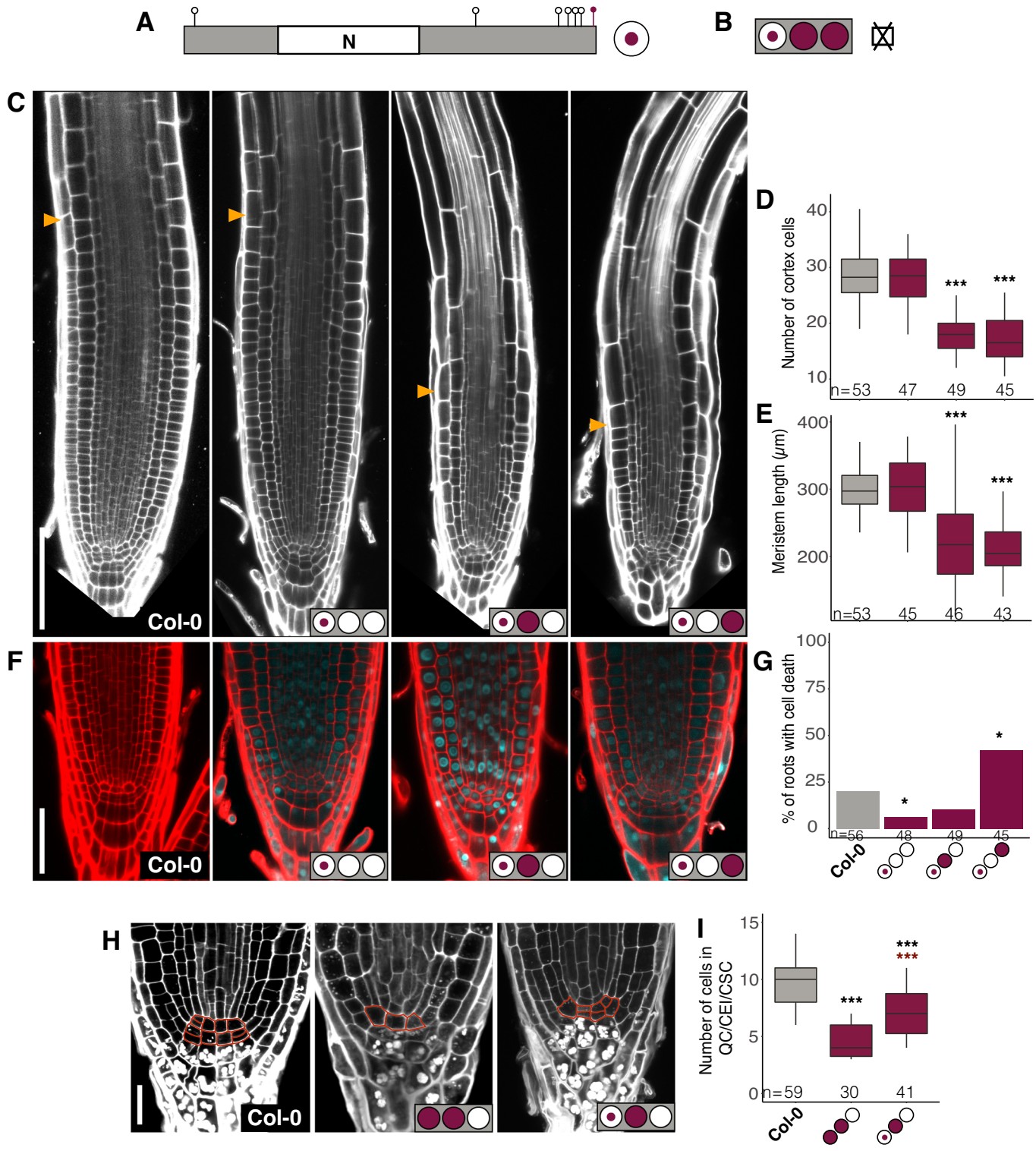

and Pocket domain sites to meristem size) contrasts with the more equal contribution of domain-specific sites to the cell death effect, as the N- and C- sites trigger cell death when combined, but not on their own (Fig. 4, compare [N-,P0,C-][11] to [N-,P0,C0][7] and [N0,P0,C-][4]). To explore the potential effects of a single specific phospho-site in one module to RBR phenotypes when combined

with defective sites in other modules, we generated a new "N" phospho-module containing a single phospho-defective site (Figs. 5A and EV1A). To this end, we mutated T406 based on conservation with the T373 residue in human pRb, reported to regulate the interaction with E2F and the LXCXE motif on its own (Burke et al, 2012; Lents et al, 2006). Combinations of the 406-

◄ **Figure 5. Combining full domain with single-site phospho-site substitutions indicates that combinatorial phosphorylation determines the function of RBR isoforms.**

(A) Schematic representation of the phospho-defective 406-module and its colored circle code. (B) The ⊠ symbol indicates that the [406-,P-,C-][10] phospho-variant is lethal. (C) Representative confocal images of PI-stained root tips of the indicated genotypes; yellow arrowheads mark the end of the meristem proliferation zone. (D, E) Box plots from (C) of meristem proliferation and size quantified as the number of cortex cells (D) and length (E) from the QC to the first rapidly elongating cortex cell. (F) Representative confocal images of PI-stained (red) root tips of the SCFP3A-tagged (cyan) indicated genotypes. (G) Bar graph from (F) showing the percentage of root tips with dead cells. (H) Representative confocal images of mPS-PI-stained root tips of the indicated genotypes showing starch granules accumulation in differentiated cells. (I) Box plot from (H) quantifying the pooled QC cells, CEI, and CSC number excluding cells with evident starch granules accumulation in the amiGO genetic background. All RBR transgenic variants are in the amiGO genetic background. Scale bars, 100 μm in (C), 50 μm in (F), 20 μm in (H). Data information: Data from two biological replicates presented as median (center line), interquartile range (box) and minima and maxima values (whiskers) (D, E, I); or as means (G); *n* denotes total number of scored roots. Fisher's test (G) or Dunett's test against Col-0 (D, E, I) and against [N-,P-,C0]12 (I), \*\*\*$P < 0.001$, \*$P < 0.05$, black asterisks indicate significant difference against Col-0; red asterisks in (I) against [N-,P-,C0][12]. Shared labels in '*x*' axis for (D, E, G).

module with the WT and phospho-defective P and C modules were analyzed for phenotype—except for [406-,P-,C-][10] that was inviable.

Similar to the fully phospho-defective N-domain, meristem size was not affected by 406- alone, but was reduced by one-third in [406-,P-,C0][6] (Fig. 5C–E). Since this effect was milder than in [N-,P-,C0][12] (Fig. 3), we conclude that T406 has an additive effect to the strong influence of the Pocket domain phosphorylation on meristem size. Notably, [406-,P0,C-][5] showed an equally strong effect (Fig. 5C–E), and even more severe than [N-,P0,C-][11] (Fig. 3) suggesting that, when combined with those in the C-region, not all phosphosites in the N-domain are additive with respect to the repressive function of RBR in meristem size maintenance.

Unlike [406-,P-,C0][6], [406-,P0,C-][5] displayed increased cell death (Fig. 5G), but to a lesser extent than its high order counterpart [N-,P0,C-][11] (Fig. 4). Since [N0,P0,C-][4] and [406-,P0,C0][1] showed full or even enhanced cell survival in the latter case (Fig. 5F,G; see Fig. 4A,B, for [N0,P0,C-][4]), we conclude that none of the phosphosites by their own, but the combination of dephosphorylated sites in the N and C regions trigger cell death, and that individual sites in the N-domain exhibit an additive effect on the phenotype penetrance.

In turn, [406-,P-,C0][6] restricted SC divisions but to a lesser extent than [N-,P-,C0][12] (Fig. 5H,I), suggesting the additive effect of N and Pocket domains phosphorylation on RBR activity, while the full complementation conveyed by [N-,P0,C0][7] and [N0,P-,C0][5] (Fig. 2), indicates that combinatorial dephosphorylation of the RBR N and Pocket domains restricts SCN activity. Unfortunately, we could not assess the effect of [406-,P0,C-][5] on SC divisions due to limited seed availability. Altogether, the phospho-defective 406 residue enhanced the activity of the P- and C-contained phosphosites to restrict cell division (to even a greater extent than the N- module when combined with C-), and triggered cell death activation only in combination with the C-terminal phospho-defective module, indicating that the phenotypic effect of an individual phospho-site depends on the phosphorylation status of the remaining ones.

## Fertility and embryogenesis are compromised in highly substituted RBR phospho-defective variants

The limited seed production of [N-T406,P0,C-][5] was also observed in [N-,P-,C0][12] and [N-,P0,C-][11]. Moreover, the few [N0,P-,C-][9] transformants we obtained that showed detectable SCFP3A fluorescence resulted in fully sterile plants (Fig. EV3A–D), highlighting the importance of phosphorylation throughout the P and C regions to sustain plant reproduction. Lack of fertilization in more

than 80% of [N-,P-,C0][12] and [N-,P0,C-][11] ovules, plus a smaller fraction of aborted seed added up to nearly 90% of sterility, regardless of the genetic background (amiGO or Col-0; Fig. EV4A). Consistently, both male and female reproductive tissues displayed cytological defects (Fig. EV4B,C). Thus, defective reproductive development results not only from reduced RBR activity as previously reported (Ebel et al, 2004; Zhao et al, 2017), but also from hyper-active isoforms, indicating that RBR is regulated by phosphorylation during gametophyte development.

Since Agrobacterium-mediated transformation occurs specifically in the female reproductive tissues (Desfeux et al, 2000), gametophytic defects may account for the lack of recovery of transgenic seedlings expressing [N-,P-,C-][16] or [406-,P-,C-][10] phosphovariants. But even if transformed ovules are fertilized, embryo lethality can also occur since RBR regulates embryonic genetic programs (Gutzat et al, 2011). To explore this possibility, we used the red fluorescent seed coat selection marker (see Fig. 1C) to select [N-,P-,C-][16] primary transformants in both amiGO and Col-0 backgrounds, and recovered all embryos from non-germinated seeds. A small fraction of embryos (~3.5%, $n = 318$) was arrested at heart- to torpedo stages and showed enlarged cells regardless the presence of endogenous RBR (Fig. 6A; Movie EV1). Some arrested embryos presented residual or absent radicles (Fig. EV3F,G), single or uneven cotyledons (Fig. EV3H,J), and signs of early differentiation like root hairs (Fig. EV3I). Conversely, we did not find any of these features in non-germinated seeds of Col-0 nor in primary transformants of a viable phospho-defective RBR variant (Figs. 6A and EV3K). Considering the phenotypic similarities in arrested embryos of [N0,P-,C-][9] transformants (Fig. EV3E; Movie EV2), our results indicate that a dominant effect of phospho-defective mutants (particularly in the P and C regions) blocks embryonic development. Altogether, defective reproduction and early developmental arrest underlie the viability loss of highly substituted phospho-defective variants, leading us to conclude that RBR multi-phosphorylation, particularly on the Pocket domain and C-terminus, is essential for plant survival.

## A point mutation in the B-pocket sub-domain rescues highly substituted phospho-defective mutants

If RBR phosphorylation disrupts its protein interactions, the dominant phenotypes of highly substituted phospho-defective RBR variants might reflect more stable protein interactions. To investigate this hypothesis, we introduced the point mutation N849F (human N757F, mouse N750F—hereafter NF), which disrupts interactions with LXCXE motif-containing proteins in

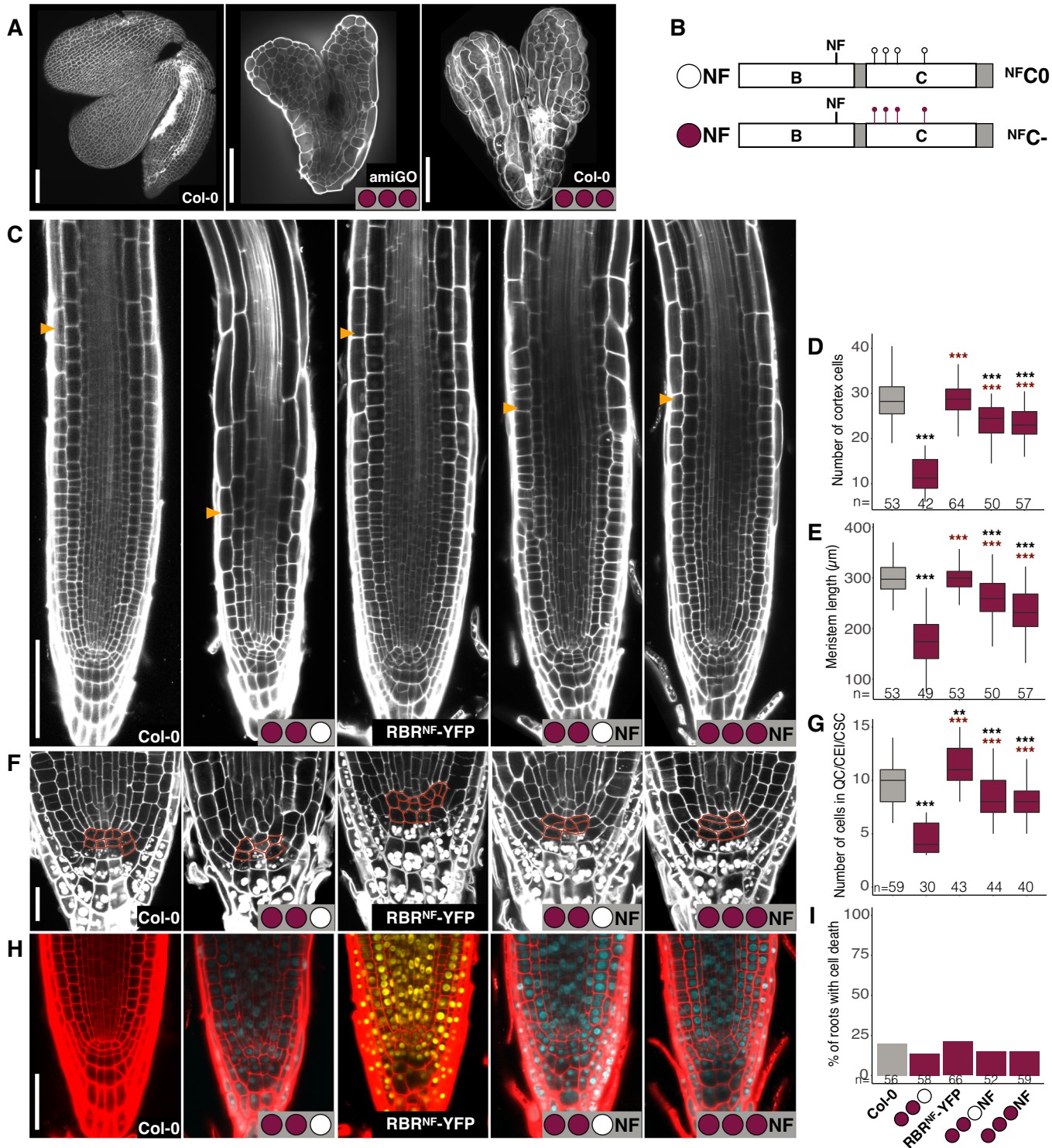

plants and animals (Bourgo et al, 2011; Chen and Wang, 2000; Cruz-Ramírez et al, 2013), into two "C" modules (Fig. 6B) to generate two new phospho-defective RBR alleles: [N-,P-,NFC0][12] and [N-,P-,NFC-][16] (Figs. 6B and EV1). Strikingly, we recovered viable plants and homozygous lines, even for the fully phospho-defective variant harboring the NF substitution.

To investigate the suppressive effect of the NF mutation, we compared the phospho-defective NF variants alongside pRBR::RBRNF:vYFP (hereafter RBRNF), with both Col-0 and [N-,P-,C0][12]. As reported previously (Cruz-Ramírez et al, 2013; Zhou et al, 2019), the RBRNF allele showed a slight overproliferation of the SCN (Fig. 6F,G). The NF mutation partially restored the meristem size phenotypes of the over-complementing

**Figure 6.   A point mutation in the B-pocket sub-domain rescues highly substituted phospho-defective mutants.**

(**A**) Confocal images of mPS-PI-stained embryos from non-germinated seeds 4 days after sowing (das) stratified for 4 days of Col-0 and primary transformants of [N-,P-,C-][16] in the genetic backgrounds amiGO and Col-0 as indicated in the back text box. (**B**) Schematic representation of the [NF]CO and [NF]C- modules and its colored circle code. (**C**) Representative confocal images of PI-stained root tips of the indicated genotypes; yellow arrowheads mark the end of the meristem proliferation zone. (**D, E**) Box plots from (**C**) of meristem proliferation and size quantified as the number of cortex cells (**D**) and length (**E**) from the QC to the first rapidly elongating cortex cell. (**F**) Representative confocal images of mPS-PI-stained root tips of the indicated genotypes. (**G**) Box plot from (**F**) quantifying the pooled QC cells, CEI, and CSC number excluding cells with evident starch granules accumulation. (**H**) Representative confocal images of PI-stained (red) root tips of the indicated SCFP3A-tagged (cyan) phosphovariants and the RBR[NF]-YFP (yellow) point mutation. (**I**) Bar graph from (**H**) showing the percentage of root tips with dead cells. All RBR transgenic variants in (**C, F, H**) are in the amiGO genetic background. Scale bars, 100 μm in (**A, C**), 20 μm in (**F**), 50 μm in (**H**). Data information: Data from two biological replicates presented as median (center line), interquartile range (box) and minima and maxima values (whiskers) in (**D, E, G**); or as means (**I**); *n* denotes total number of scored roots. Fisher's test (**I**) or Dunett's test against Col-0 and against [N-,P-,C0][12] (**D, E, G**), ***$P < 0.001$, **$P < 0.01$, black asterisks indicate significant difference against Col-0; red asterisks, against [N-,P-,C0][12]. Col-0 and [N-,P-,C0][12] values are the same as in Fig. 5 as experiments were performed in parallel sharing these controls. Shared labels in 'x' axis for (**D, E, G, I**).

[N-,P-,C0][12] variant (Fig. 3 vs Fig. 6C–E); similarly, SCN differentiation was also partially rescued (Fig. 2 vs Fig. 6F,G). Moreover, the fully phospho-defective [N-,P-,[NF]C-][16] variant showed little, if any, phenotypic variation compared to [N-,P-,[NF]C0][12] (Fig. 6C–I). Even non-germinating [N-,P-,[NF]C-][16] primary transformants showed more advanced development than [N-,P-,C-][16] arrested embryos (Fig. 6A vs EV3K). Thus, the partial rescue of highly substituted phospho-defective variants phenotypes by the NF mutation suggests that RBR-LXCXE protein interactions constitute a predominant component of RBR-mediated developmental processes regulated by phosphorylation.

### RBR protein–protein interactions with transcriptional regulators are differentially regulated by phosphorylation

To investigate if RBR direct protein interactions reflect our phenotypic observations, we performed Yeast Two-hybrid (Y2H) screenings of the Arabidopsis pEXP22-TF collection (Pruneda-Paz et al, 2014) using eight highly substituted RBR phosphovariants, RBR[NF], and wild-type RBR as baits. Although the total number of interactors in the screenings summed up to 28 proteins (Appendix Table S1), co-transformation and stringent selection of interacting baits and prays confirmed 14 interactors with varying binding properties, some of which were previously reported and others were unknown (Appendix Fig. S1). Several interactors are involved in responses to stress, four of which (DREB2D, GBF4, NAC090, NAC044) interacted strongly with all RBR phosphovariants, pointing to a phosphorylation-independent role of RBR as an integrator of environmental inputs. Conversely, TFs related to cell proliferation and development (TCX6/7, E2FC, XND1, TCP3) showed weaker or no interaction with most phospho-mimetic variants. Taken together with our phenotypic analysis, these results indicate that RBR protein structure is functional despite the multiple mutations, and that phospho-site substitutions work as expected in both of their defective and mimetic versions. Two interactors revealed special properties. On the one hand, ARIA interacted strongly and exclusively with RBR variants containing an intact N-domain, suggesting that this protein docks closely to, or even on phosphosites at the N-domain. On the other hand, GRF5 showed strong but variable binding properties to RBR, since it interacted with all RBR variants in at least one replicate, but consistently only with Wt RBR, [N-,P0,C-][0], and [N-,P-,C-][16] (Appendix Fig. S1B), suggesting that other factors might be required to stabilize the RBR-GRF5 complex, like the co-activator GRF-INTERACTING FACTOR1/ANGUSTIFOLIA3 (GIF1/AN3).

### Phosphorylation-regulated functions of RBR are largely mediated by the interaction with members of the DREAM complex

We paid special attention to the cysteine-rich proteins TCX6 and TCX7, members of the multimeric DREAM complex, a conserved eukaryotic cell cycle regulator recently described in plants (Kobayashi et al, 2015; Lang et al, 2021; Ning et al, 2020). TCX6/7 displayed a decreased affinity for phospho-mimetic RBR variants and no interaction with RBR[NF]. Together with TCX5, TCX6 and TCX7 contain a conserved LXCXE motif responsible for the interaction with RBR that is absent in the remaining TCX family members (Fig. EV5A–C); (Lang et al, 2021). Since TCX7 expression was undetectable in all tissues analyzed (Andersen et al, 2007), and the double, but not the single mutants of tcx5 and tcx6 exhibit phenotypes in other plant parts (Ning et al, 2020), we analyzed the root meristem of the tcx5/6 double mutant. The tcx5/6 mutant exhibited increased meristem size, cell proliferation, and cell death (Fig. EV5D–H), similar to the phenotypes of the amiGO root tips, suggesting that the RBR-TCX5/6 interaction is relevant for RBR function. Thus, we transformed the dominant lethal phospho-defective [N-,P-,C-][16] variant of RBR in the tcx5/6 mutant background. Surprisingly, we recovered 16 independent primary transformants with detectable CFP3A nuclear signal, indicating that the absence of TCX5/6 proteins is enough to circumvent the dominant lethality of such a variant. Nevertheless, only one transformant line generated T2 seed, indicating that the suppression of the [N-,P-,C-][16] variant by tcx5/6 is weaker than that observed in [N-,P-,[NF]C-][16]. Accordingly, the phenotypes of the T2 tcx5/6;[N-,P-,C-][16] line are stronger than those observed for [N-,P-,[NF]C-][16] (compare Fig. EV5D with Fig. 6), despite the weaker SCFP3A intensity of the former. Altogether, we concluded that the roles of RBR regulated by phosphorylation and mediated by its LXCXE-binding properties, partially depend on the interaction with the DREAM complex members of the TCX5/6/7 clade.

## Discussion

RBR is associated with multiple and complex roles in development, and it has been unclear how a single protein can carry out a wide range of functions through regulated interactions with different protein partners. Here, we have explored the separability of roles for plant RBR phosphorylation, which has emerged as a prominent

regulatory mechanism of RBR in the multitude of RB protein functions described so far. Taken together, our results support the notion that additive phosphorylation fine-tunes RBR activity and function, and combinatorial phosphorylation provides the potential for separating its various roles. While phospho-defective variants often showed hyper-active dominant effects and over-complementation of 'amiGO' plants (Figs. 2, 3, 4C,D, 5, 6A, EV3, and EV4A), the phenotypic strength of phospho-mimetic variants ranged between those observed for Col-0 and amiGO (Figs. 2–4), which supports the prevailing conception that phosphorylation reduces RBR activity. But three additions to this generic idea are to be made: phosphorylation events on RBR (1) are independent of each other, (2) unequally contribute to RBR activity, and (3) disentangle RBR functions.

## RBR phosphosites are independent of each other

We observed an additive effect in the phenotypic strength as the number of mutated phosphosites was increased in RBR variants. Together with the phenotypic differences between full phospho-site variants ([N-,P-,C-][16] and [N+,P+,C+][16]) and all single phospho-module combinations, our findings exclude a nucleation mechanism for RBR hyper-phosphorylation, and demonstrate that phosphorylation events on RBR are independent of each other.

## Uneven contribution of phosphosites to RBR activity regulation

Unlike [N-,P-,C0][12] and [N-,P0,C-][11], the less substituted phospho-defective variant [N0,P-,C-][9] was lethal. Therefore, the phosphorylatable Pocket domain and C-region, but not the N-region of RBR protein, were sufficient to sustain plant growth and viability despite bearing less phosphosites. This excludes a simple phospho-counting mechanism as the primary mode of RBR regulation. Moreover, phosphorylation within the Pocket domain and the C-region markedly influenced the proliferative activity of the meristem and SCN, respectively, whereas phosphorylation of the N-domain seemed unimportant on its own (Figs. 2 and 3). Accordingly, phosphorylation within the Pocket domain and C-region of pRb regulate E2F and LXCXE motif binding (Burke et al, 2010; Knudsen and Wang, 1997), while phosphorylating the N-domain becomes relevant in response to stress (Gubern et al, 2016). Future research should unveil the functions of RBR N-domain phosphorylation during plant stress responses.

## RBR functions are separable by phosphorylation

While the phospho-defective variant [N0,P-,C0][5] over-complemented root meristem size but not SCN division, all three double phospho-mimetic modules combinations displayed overproliferation of the SCN but not of transit amplifying cells. Notably, several phospho-mimetic variants that failed to restrict SCN activity, complemented the cell death phenotype. On the other hand, [N-,P0,C-][11] and [N-T406,P0,C-][5] repressed cell division and frequently displayed dead cells, whereas [N-,P-,C0][12] and [N-T406,P-,C0][6] blocked meristematic and stem cell division without inducing cell death (Figs. 2–4). Contrary to previously observed pleiotropic effects in knock-out or altered expression approaches, our findings revealed the capacity of RBR to

regulate independently cell division, differentiation and survival according to its phosphorylation state.

We observed increased cell death in roots of both phospho-mimetic and phospho-defective variants. Similarly, apoptotic stimuli can promote phosphorylation as well as dephosphorylation of pRb (De Leon et al, 2008; Nath et al, 2003); pRb in turn, can either promote or inhibit apoptosis (Antonucci et al, 2014; Goodrich, 2006; Ianari et al, 2010), a fate decision largely mediated by its phosphorylation state (Antonucci et al, 2014; Egger et al, 2016; Gubern et al, 2016; Lee et al, 2018; De Leon et al, 2008; Nath et al, 2003). Which particular phosphorylation state is preferred, and what outcome it takes might depend on circumstances. In plants, biotrophic attackers promote RBR hyper-phosphorylation to trigger immunity-related Programmed Cell Death (PCD) (Wang et al, 2014a), correlating with our fully phospho-mimetic variant. We speculate that rapid immune responses to pathogen attack, sensed and signaled by phospho-relay cascades, prioritize an urgently required activation of PCD to avoid infection spread, whereas developmental PCD might involve a more accurate mechanism, reflected by the combinatorial specificity of our phospho-defective variants triggering cell death and inhibiting cell division (Figs. 4 and 5L). Thus, hyper-phosphorylation of RBR may well act "quick and dirty" to counteract stresses, while combinatorial phosphorylation entails a timely coordination of cell fate decisions.

Great endeavors in the late 90 s utilized systematic mutagenesis to understand the functional nature of pRb phosphorylation (Barrientes et al, 2000; Brown et al, 1999; Knudsen and Wang, 1997, 1996; Knudsen et al, 1999), pointing to a combinatorial role in pRb regulation (Munro et al, 2012; Rubin, 2013). But this notion has been challenged in recent years based on a report where pRb was found in only three states in cellular lines: un-phosphorylated, hyper-phosphorylated and mono-phosphorylated (Narasimha et al, 2014). A more recent report (Sanidas et al, 2019), recapitulated the concept of the phosphorylation code, but focused on mono-phosphorylated isoforms. In that study, the authors found that more than one-third of the pRb interactome (175 out of 438 proteins) bind neither un-phosphorylated nor any of the 14 mono-phosphorylated variants, and assumed that these interactions correspond to hyper-phosphorylated pRb (Sanidas et al, 2019). Since the phenotypes of our fully phospho-mimetic variant suggest that hyper-phosphorylated RBR is mostly inactive, we believe more evidence is needed before the possibility of intermediate phosphorylation states of RB proteins is rejected. In particular, investigation of pRb phosphorylation states in the whole-organism context remains a future challenge.

To our knowledge, this is the first systematic study of RBR phosphovariants in a full multicellular organismal context. We did not determine the existence of intermediate phosphorylation RBR isoforms in planta, limiting the reach of our observations. However, our "artificial" phosphovariants recapitulated functional outcomes of RBR, implying its potential to "interpret" a combinatorial multisite phosphorylation modulated by additive effects. We are aware that not all putative phosphosites have been confirmed in vivo, but all phospho-defective variants showed at least a mild and additive effect, implying that at least the majority of phosphosites are functional. We also noticed that in very few cases, like [N-,P-,C0][12] and [N+,P+,C0][12] effect in the SCN proliferation, the corresponding phospho-defective and phospho-mimetic variants lead to opposite effects. Due to the fertility defects

precluding normal expression of the [N0,P-,C-][9] and [N-,P0,C-][11] variants, opposite effects with their corresponding phospho-mimetic variants cannot be excluded. However, opposite phenotypes are not a necessary outcome of our approach, as they are expected only if the corresponding phosphosites are phosphorylated and dephosphorylated at similar rates in the WT protein. Similarly, the non-mutated sites in each variant contribute to the phenotypic outcome but their phosphorylation status is unknown.

Two shortcomings in our approach are the potential effect of residual RBR in the amiGO genetic background, and the static nature of the phosphorylation substitutions—contrasting with the dynamics entailed by phosphorylation-dependent regulation of a multifunctional protein, achieved by concerted action of CDKs and phosphatases. The former issue could be addressed by postembryonic gene editing to generate a postembryonic *rbr* null background; the latter would require thorough characterization of the endogenous phosphorylation isoforms of RBR within its diverse spatio-temporal contexts. In this regard, single-cell and single-molecule approaches promise an exciting future for RB protein biology.

Our results support a combinatorial mode for RBR regulation by phosphorylation and suggest that distinct CYC-CDKs complexes target RBR phosphosites with distinct affinities as is the case for CYCDs on pRb (Paternot et al, 2006). For instance, CYCD3;1 overexpression leads to overproliferation in the SCN area without inducing cell death (Horvath et al, 2017), resembling the phospho-mimetic variants [N0,P+,C+][9], [N+,P0,C+][11], and [N+,P+,C0][11], indicating that CYCD3 might direct phosphorylation of many but not all phosphosites. Moreover, the T406 site only showed prominent effects in combination with other residues, and together with S911 it is preferentially phosphorylated by CYCA3;4, whose overexpression results in RBR-associated phenotypes (Willems et al, 2020); and CYCD6;1, a developmental and stress-responding gene (Bertolotti et al, 2020; Cruz-Ramírez et al, 2012; Zhou et al, 2019), drives the kinase activity of CDKB1 to the RBR pocket domain (Cruz-Ramírez et al, 2012). Taken together with the distinct expression patterns of CYC genes (Collins et al, 2012; Menges et al, 2005), the intricate regulation of CDK activity (Sanz et al, 2011), and their substrate-specificity, these observations indicate a complex mechanism to orchestrate RBR multiple functions by combinatorial phosphorylation during the plant life cycle, posing new challenges to our understanding of RBR networks and its regulation by a diversity of CYC-CDK complexes.

On the same track, RBR protein interactions mediated by the B-pocket constitute a major component of phosphorylation-mediated functions of RBR (Fig. 6), pointing out to proteins containing an LXCXE motif. Noteworthy, RB interactions with LXCXE-containing proteins, both in plants and animals, play prominent roles in sustaining differentiation and growth arrest decisions (Chen and Wang, 2000; Cruz-Ramírez et al, 2012; Matos et al, 2014), and to withstand stressful growth conditions (Bourgo et al, 2011; Collins et al, 2015; Cruz-Ramírez et al, 2013; Zhou et al, 2019). The significant rescue of all phenotypes associated with highly substituted phospho-defective variants by the NF mutation reveals the vast importance of LXCXE protein interactions, but at the same time, it points to an LXCXE-independent component in the phosphorylation-dependent functions of RBR. We showed that RBR interactions with the LXCXE-containing proteins of the TCX5/6/7 clade and with the E2FC proteins, both members of the DREAM complex, are regulated by phosphorylation as they interacted notably stronger with phospho-defective variants than with phospho-mimetics. As the NF substitution conferred stronger suppression of the fully phospho-defective RBR than the *tcx5/6* genetic background, we conclude that several but not all phosphorylation-regulated roles of RBR are intimately linked to the DREAM complex.

Our Y2H analysis suggests new features of RBR previously unknown. First, several interactors are linked to stress and/or environmental responses, like DREB2D to dehydration, high salinity, and heat stress (Liu et al, 1998; Chen et al, 2010; Nakashima et al, 2000); GBF4 to cold and dehydration (Lu et al, 1996; Menkens and Cashmore, 1994); NAC090 to reactive oxygen species, salicylic acid responses and sound vibrations that elicit defense hormones (Kim et al, 2018; Ghosh et al, 2016), the latter being a proposed mechanism to perceive herbivore chewing (Appel and Cocroft, 2014); NAC044 to DNA damage and heat stress (Takahashi et al, 2019); ARIA to the stress hormone ABA (Kim et al, 2004); and POB1 to light, vernalization, and susceptibility to pathogens (Christians et al, 2012; Hu et al, 2014; Pogoda et al, 2020); and interestingly, POB1 regulates cell death in tobacco (Orosa et al, 2017) and interacted almost exclusively with the cell death-inducing variant [N-,P0,C-][11]. A subset of these interactors, bind RBR independently of its phosphorylation state, suggesting that hyper-phosphorylation does not inactivate RBR completely. Second, the NF mutation disrupted the interaction with TCP3, but only NAC044, and TCX6/7 interact with RBR in a LXCXE-dependent manner (Lang et al, 2021), (Appendix Fig. S1; Fig. EV5C) moreover, TCP3 and TCX6/7 interactions with RBR are weakened by the phospho-mimetic mutations. Therefore, phosphorylation may not necessarily block the LXCXE-binding site in RBR as previously thought (Gutzat et al, 2012), and not all proteins binding to this site actually contain an LXCXE motif. Third, RBR-ARIA interaction seems controlled by local rather than global changes in the vicinity of phosphosites within the N-domain, suggesting a novel mechanism of PPI-regulation by RBR phosphorylation.

Altogether, we have taken first steps to understand the combinatorial nature of RBR phosphorylation. Our biochemical and phenotypic analysis suggests that the integrative functions of RBR (Harashima and Sugimoto, 2016) seem to rely on both phosphorylation-regulated and phosphorylation-independent interactions with nuclear proteins. Combinatorial phosphorylation of RBR is essential for developmental processes like (stem) cell division, cell death and differentiation, but seemingly unimportant for binding to several stress-related proteins. Finally, our RBR phosphovariants collections and combinable phospho-modules are a valuable resource for future research. On the one hand, the characterization of in vivo-phosphorylated sites under environmentally varying conditions may guide the choices to expand the collection. On the other hand, using the collection for cell-type specific, high-throughput experiments and studies on CYC-CDK specificities could help to understand RBR networks throughout development and stress responses.

# Methods

## Accession numbers

RETINOBLASTOMA-RELATED (RBR), AT3G12280; G-BOX BINDING FACTOR 4 (GBF4), AT1G03970; E2FC, AT1G47870; Teosinte branched1 Cycloidea1 and PCNA factor 3 (TCP3), AT1G53230; DRE-

BINDING PROTEIN 2D (DREB2D), AT1G75490; Tesmin/TSO1-like CXC domain-containing protein 6 (TCX6), AT2G20110; TSL-KINASE INTERACTING PROTEIN 1 (TKI1), AT2G36960; NAC DOMAIN-CONTAINING PROTEIN 44 (NAC044), AT3G01600; GROWTH REGULATING FACTOR 5 (GRF5), AT3G13960; POZ/BTB CONTAINING-PROTEIN 1 (POB1), AT3G61600; HD-like, AT4G03250; ARM REPEAT PROTEIN INTERACTING WITH ABF2 (ARIA), AT5G19330; NAC DOMAIN-CONTAINING PROTEIN 90 (NAC090), AT5G22380; Tesmin/TSO1-like CXC domain-containing protein 7 (TCX7), AT5G25790; XYLEM NAC DOMAIN 1 (XND1), AT5G64530.

## Plant material and growth conditions

*Arabidopsis thaliana* ecotype Col-0 was used as wild-type control. Unless otherwise noticed, amiGO-RBR (Cruz-Ramírez et al, 2013) was used as genetic background for transgenic plants. Seeds were fume-sterilized in a sealed container with 100 ml bleach and 3 ml of 37% hydrochloric acid for 3–5 h; then suspended in 0.1% agarose, stratified for 2 days (4 days for arrested embryos) at 4 °C in darkness, plated on 0.5× Murashige and Skoog (MS) plus vitamins, 1% sucrose, 0.5 g/l 2-(N-morpholino) ethanesulfonic acid (MES) at pH 5.8, and 0.8% plant agar, and grown vertically for 6 days (4 days for arrested embryos) at 22 °C with a 16 h light/8 h dark cycle. For cytological analysis of gametophytes, seedlings were transplanted to soil and grown until the reproductive stage.

## RBR phosphovariants plasmid construction and Plant transformation and transgenic selection

The Golden Gate modular cloning (MoClo) system for plants (Engler et al, 2014) and adequate primer pairs (Appendix Table S3) were used to generate all phosphovariants. A detailed description of phosphovariants cloning is offered in Appendix Supplementary Methods. Level 2 constructs were transformed in homozygous amiGO plants by flower dip method. Primary tansformants were selected for fluorescent red seed coats under a fluorescence microscope; at least 16 primary transformants were visualized with confocal laser scanning microscopy (CLSM) and selected for the presence of nuclear SCFP3A signal. At least two independent lines were taken to homozygous T3 generation for phenotyping.

## Microscopy

A 10 µg/mL Propidium Iodide (PI) staining solution was used for whole-mount visualization of live roots with CLSM using a Zeiss LSM 710 system as described in (Zhou et al, 2019). For arrested embryos, seed coats were removed as described in (Lee and Lopez-Molina, 2013); modified pseudo-Schiff PI (mPS-PI) staining of roots and embryos was performed as described in (Zhou et al, 2019). Images were taken with ZEN 2012 software (Zeiss) and processed with ImageJ software, using the Stitching Plug-in for multiple images. Brightness and contrast of the final figures was enhanced to the exact same values except for Figs. EV3B,C and EV5E, where CLSM and brightness/contrast parameters were maximized due to the very weak SCFP3A signal. DIC images of gametophytes were obtained with a Nomarsky illumination Leica DRM system.

## Phenotypic analysis

At least two independent transgenic lines for each genotype were analyzed in at least two independent occasions. For the three phenotypes analyzed (meristem size, stem cell proliferation, and cell death), independent replicates of each line were compared among them; and independent lines of the same genotype we compared among them, with no significant difference. Each line was compared with the Col-0 control, obtaining similar results for each line of the same genotype. Only one line is presented for each genotype. Quantification of the root meristem size was done by imaging the median longitudinal section of the root tip and averaging the number of cortex cells from the QC to the first rapidly elongating cell in the two visible cortex layers; and by measuring and averaging the distance spanned by these cells. Cell death was quantified as the percentage of root tips presenting dead cells as visualized with PI and scanning throughout the Z-axis. Stem cells were visualized by mPS-PI staining and imaging the median longitudinal section of the root tip. Statistical analysis was performed by a one-way ANOVA followed by Dunnett's multiple comparisons test, or Chi-square, followed by Fisher's exact test using GraphPad Prism version 5.0.0 for Windows (GraphPad Software, San Diego, California USA www.graphpad.com) which retrieves the significance of each comparison in terms of $P$ values < 0.05, 0.01, 0.001 and so on. Comparisons to assess the differences between replicates of the same genotype were performed before comparing different genotypes. All replicates from the same genotype reposted in this study showed no statistical difference. Histological analysis of female gametophytes was performed with one line per genotype as described previously (Demesa-Arévalo and Vielle-Calzada, 2013).

All constructs and plant lines are available upon request.

## Data availability

This study includes no data deposited in external repositories.

The source data of this paper are collected in the following database record: biostudies:S-SCDT-10_1038-S44318-024-00282-3.

## Peer review information

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

## Acknowledgements

The authors thank Prof. Jim Murray (Cardiff University) for his valuable suggestions on an early version of this manuscript; Dr. Xin-Jian He (National Institute of Biological Sciences, China) for sharing their *tcx5/tcx6* mutant line; Dr. Sara Díaz-Triviño (Wageningen University) for sharing the unpublished RBR$^{NF}$-YFP line, Dr. Yessica Alina Rodríguez-Rosales (Radboud UMC) for her support with statistical analysis and R plots, Jonathan Matthew Samson, Margot Smith and Dr. Renze Heidstra for technical support and advice. JZZ was funded by Consejo Nacional de Ciencia y Tecnología (CONACyT, Mexico, 383871).

## Author contributions

**Jorge Zamora-Zaragoza**: Conceptualization; Resources; Data curation; Formal analysis; Funding acquisition; Validation; Investigation; Visualization; Methodology; Writing—original draft; Project administration; Writing—review and editing. **Katinka Klap**: Investigation; Methodology. **Jaheli Sánchez-Pérez**: Data curation; Investigation; Visualization. **Jean-Philippe Vielle-Calzada**: Supervision; Visualization; Methodology. **Viola Willemsen**: Formal analysis; Validation; Investigation; Visualization; Methodology. **Ben Scheres**: Conceptualization; Resources; Supervision; Writing—review and editing.

Source data underlying figure panels in this paper may have individual authorship assigned. Where available, figure panel/source data authorship is listed in the following database record: biostudies:S-SCDT-10_1038-S44318-024-00282-3.

## Disclosure and competing interests statement

The authors declare no competing interests.

# Expanded View Figures

**Figure EV1.   Modules and RBR-phosphovariants.**

(**A**) Schematic list of all modules cloned in Level -1 (vector pAGM1311) of the GoldenGate MoClo system. Text nomenclature and colored circles and their relative position within the gray background box indicate the phosphorylation state and position of each module within the full-length CDS of phosphovariants according to Fig. 1. Note that in modules "406" only the Thr406 residue is mutated, and in modules "$^{N F}$C" the Asn849 within the LXCXE-binding cleft of the B-pocket sub-domain is mutated to Phe. (**B**) Schematic list of all phosphovariants generated. All variants listed exist as Level 0 (vector pAGM1287), Level 1 (vector pICH47742; with RBR promotor, SCFP3A CDS and NOS terminator), and Level 2 (vector pAGM4723; with FAST-R selection cassette in position 1) constructs, and as transgenic seed, except for those marked with the symbol ⊚ on the left-most column, which were either not viable or not transformed and thus, only the plasmids are available. Note that [406-,P0,C-]$^5$ is marked with double circle because reduced fertility hindered propagation and all seed was used for the experiments reported in Fig. 5.

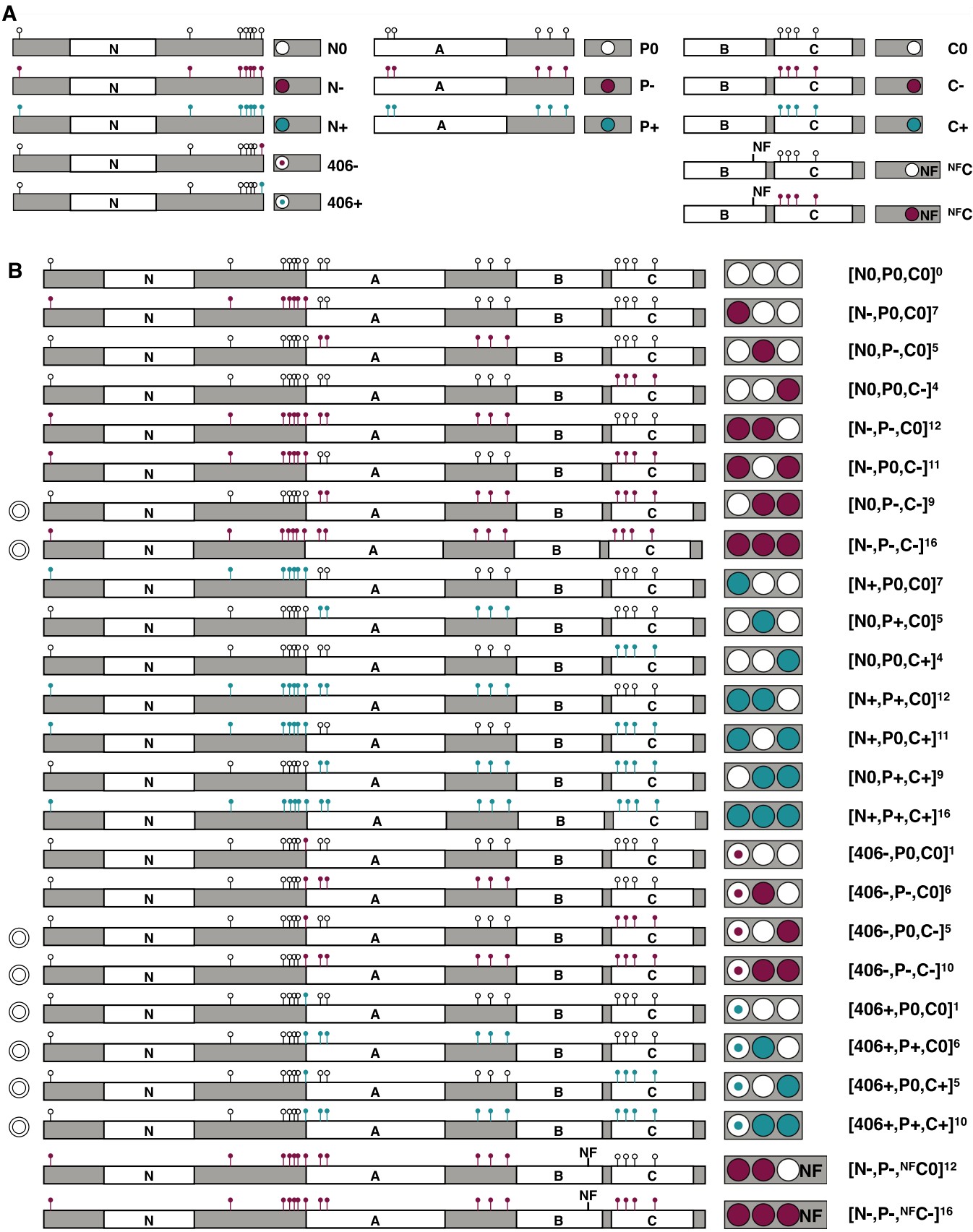

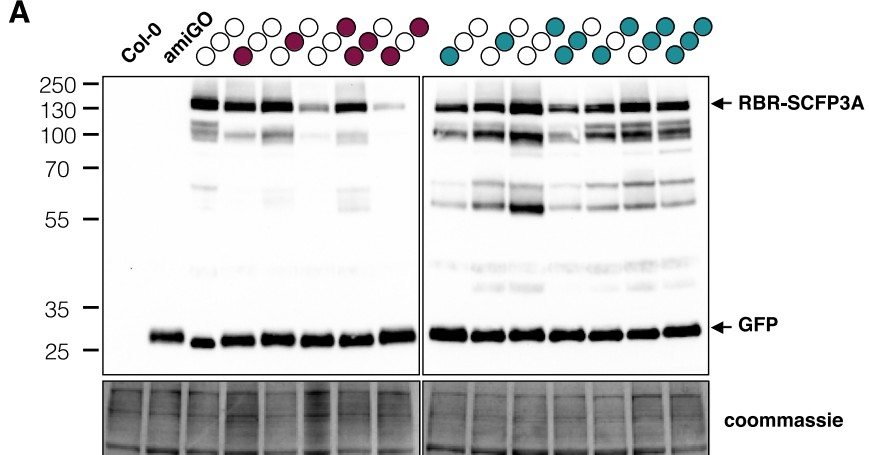

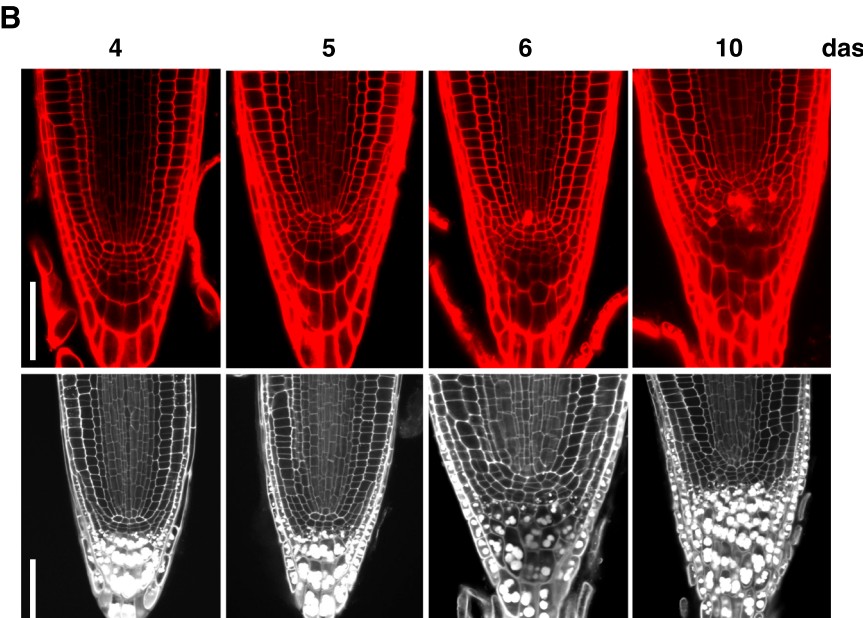

**Figure EV2. Characterization of amiGO-RBR phenotype penetrance and RBR phosphovariants protein accumulation.**

(A) Western blot analysis of RBR phosphorylation variants. The anti-GFP antibody was used to detect the SCFP3A-tagged RBR transgenic variants and the free GFP co-expressed in all lines containing the amiGO construct. (B) Confocal images of amiGO-RBR root tips by 4, 5, 6 and 10 days after sowing (das). Top panels, mPS-PI staining; bottom panels, PI staining. Red spots in the SCN area correspond to cell death, as PI selectively stains dead cells. Note that by 5 das we detected roots with very weak or no amiGO-associated phenotypes. By 6das, cell death and SCN extra divisions were evident in all roots. Scale bars, 50 μm.

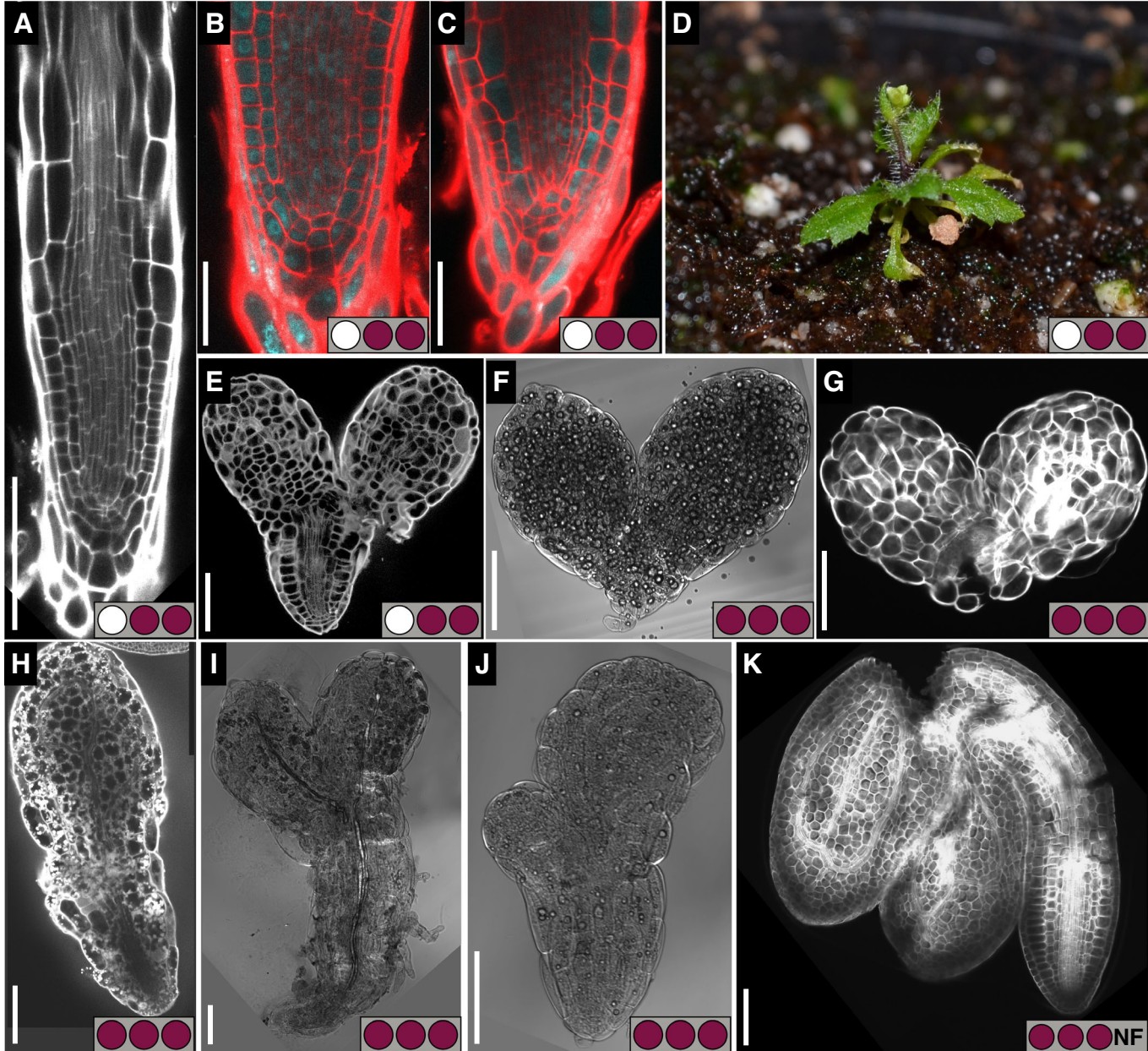

**Figure EV3.   Primary transformants of lethal phospho-defective RBR variants.**

(A–C) Confocal images of PI-stained root tips. Max power and gain for CLSM settings and image brightness and contrast in (B, C) were set to in order to visualize SCFP3A signal from [N0,P-,C-][9] expression (Cyan). (D) 3 week old seedling. (E–K) Confocal images of mPS-PI-stained (E, G, H, K), and transmitted light images (F, I, J) of embryos from non-germinated seeds 4 das stratified for 4 days. Genotypes: [N0,P-,C-][9] (A–E), [N-,P-,C-][16] (F–J), [N-,P-[NF],C-][16] (K); Genetic background: Col-0 (A, B, G, K), amiGO (C–F, H–J). Scale bars, 100 µm in (A, E–K), 50 µm in (B, C).

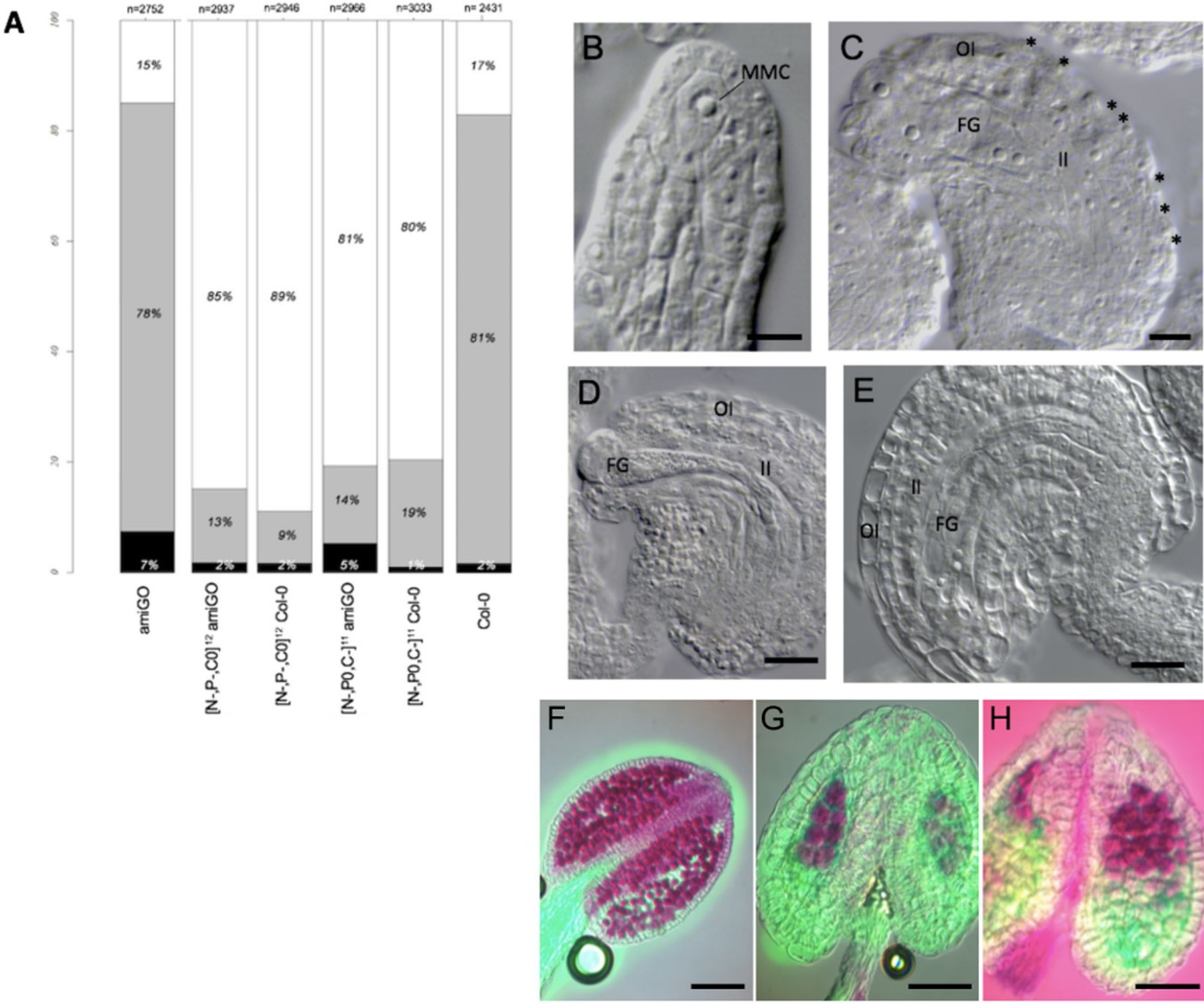

**Figure EV4. Fertility and embryogenesis are compromised in the highly substituted RBR phospho-defective variants [N-,P-,C0]¹² and [N-,P0,C-]¹¹.**

(A) Sterility analysis quantified as percentage of non-fertilized ovules, aborted seed and mature seed. N denotes total number of scored ovules and seed. (B–E) DIC images of ovule development of [N-,P0,C-]¹¹ (B, E) and [N-,P-,C0]¹² (C, D). Ovule primordia with one normal precursor cell (B). Incomplete integument development results in abnormally exposed embryo sac (C–E). Scale bars 10 µm in (B), 20 µm in (C–E). (F–H) Alexander staining of Col-0 (F), [N-,P-,C0]¹² (G), and [N-,P0,C-]¹¹ (H) anthers showing viable pollen grains in fuchsia. Scale bars 100 µm.

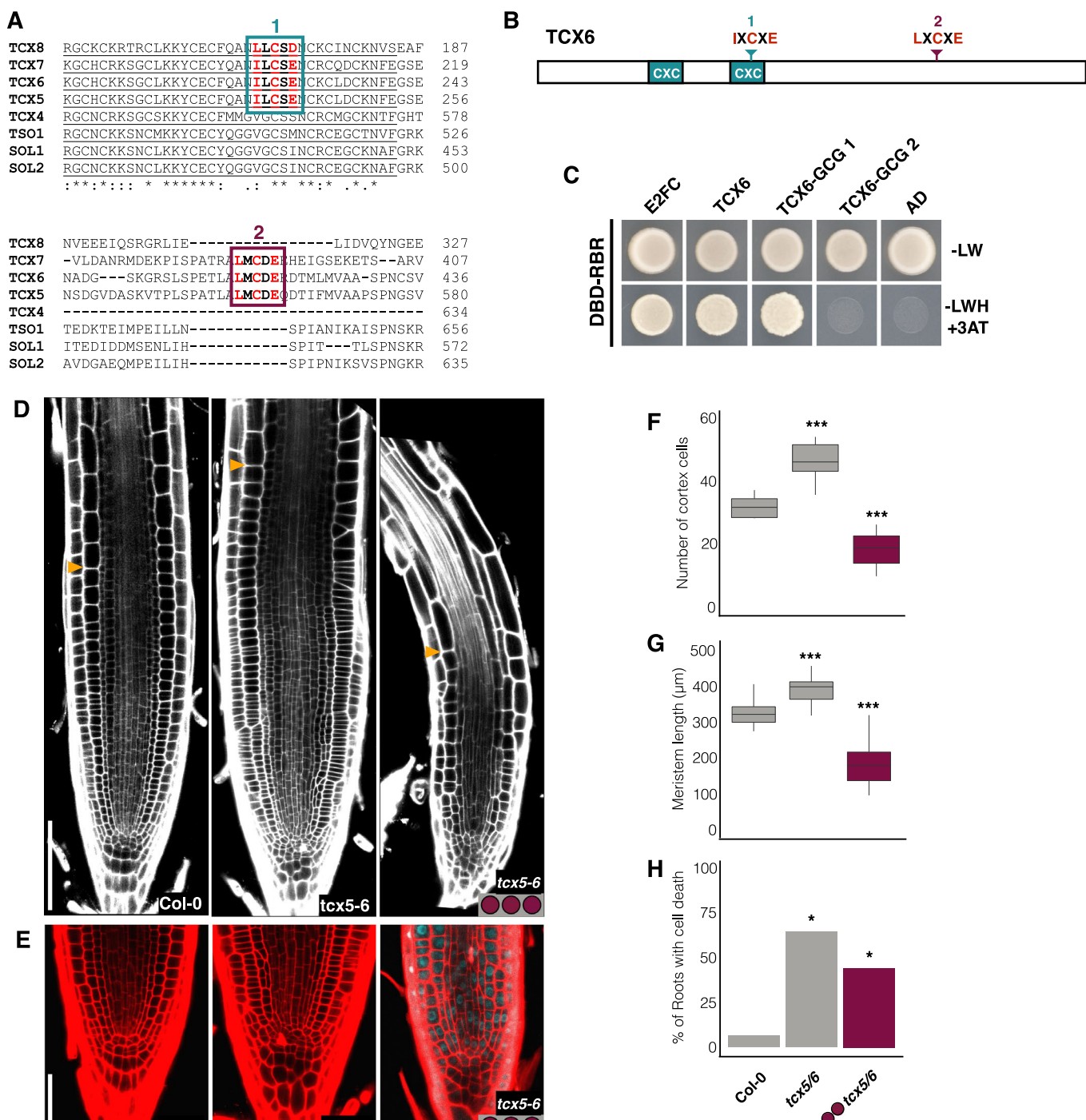

◀ **Figure EV5. Phosphorylation-regulated functions of RBR are largely mediated by the interaction with members of the DREAM complex.**

(A) Clustal omega multiple sequence alignment fragments of Arabidopsis TCX proteins showing LXCXE-like motifs (red) within the conserved CXC domain (underlined) and LXCXE motifs (in green) within a less conserved region; asterisks and dots indicate identical and similar residues, respectively. (B) Schematic representation of TCX6 protein organization as predicted by Pfam server (https://pfam.xfam.org/), showing the relative positions of the cysteine-rich domains (cxc) and LXCXE and LXCXE-like motifs. (C). Yeast two-hybrid analysis showing that RBR interacts with TCX6 and a TCX mutated on the LXCXE-like motif '1' within the CXC domain, but not with TCX6 mutated on the canonical LXCXE motif '2'. E2FC is positive control, and empty pDEST22 vector is negative control. Co-transformed yeast dropped on SD –LW to select transformants, and on SD -LWH + 1.0 mM 3AT to select interactions. (D, E) Representative confocal images of PI-stained root tips of the indicated genotypes; yellow arrowheads mark the end of the meristem proliferation zone. The CLSM settings for detecting SCFP3A in the *tcx5/6*;[N-,P-,C-][16] were identical than those for all other phosphovariants, but brightness and contrast were enhanced to visualize the nuclear signal due to the low fluorescence intensity. Red spots in the SCN area correspond to cell death, as PI selectively stains dead cells. Scale bars, 100 μm in (D), 50 μm in (E). (F, G) Box plots from (D) of meristem proliferation and size quantified as the number of cortex cells (F) and length (G) from the QC to the first rapidly elongating cortex cell. (H) Bar graph from (E) showing the percentage of root tips with dead cells. Data information: Data from one biological replicate presented as median (center line), interquartile range (box) and minina and maxima values (whiskers) in (F, G); or as means in (H); $n > 15$ in (F, G, H). Wilcoxon test against Col-0, ***$P < 0.001$ in (F, G), Chi-square, *$P < 0.05$ in (H).

