## [Peer Review File · The EMBO Journal]

Developmental cues are encoded by the combinatorial phosphorylation of Arabidopsis RETINOBLASTOMA-RELATED protein RBR1

Jorge Zamora-Zaragoza, Katinka Klap, Jaheli Sánchez-Pérez, Jean-Philippe Vielle-Calzada, Viola Willemsen, and Ben Scheres

Corresponding author(s): Ben Scheres (ben.scheres@wur.nl)

Review Timeline:

Submission Date:	22nd Dec 21
Editorial Decision:	18th Feb 22
Revision Received:	1st Jun 24
Editorial Decision:	9th Aug 24
Revision Received:	29th Aug 24
Accepted:	27th Sep 24

Editor: William Teale

Transaction Report:

Dear Prof. Scheres,

Thank you again for the submission of your manuscript entitled "A phosphorylation code regulates the multi-functional protein RETINOBLASTOMA-RELATED1 in Arabidopsis thaliana". We have now received the referees' reports, which I have copied to the bottom of this message. I would also like to apologise for the unusually long time it has taken to collect these reports.

As you can see, the reports are generally supportive of your manuscript, but express some significant concerns. All referees agree that, at its heart, the work is based on a technically accomplished collection of experiments. They also state unambiguously that the manuscript is timely and the topic is important. However, the feedback was not unambiguously positive. Firstly, two of the referees were unable to understand, conceptually, the code to which the manuscript's title refers. This has had a negative impact on the extent to which your findings on the mechanism of RBR regulation and function can be transferred. Secondly, all referees were concerned that some of the data you report need additional orthologous support before they can be published in EMBO Journal.

On balance though, I would like to invite you to address the comments of all referees in a revised version of the manuscript. I encourage you to include in this revised manuscript the characterization with the antibody suggested by reviewer 2 (point 1), and the expanded phenotypic characterization suggested by reviewer 1 (point 1). As you will see, two expert referees struggled to extract the significance of your findings surrounding T406 phosphorylation. I therefore urge you to re-organise the manuscript to focus the readers' attention on the context for your decision to focus your work here. I recommend we talk next week by Zoom to discuss the referees' comments further; if you would like to do this, please suggest a couple of time slots. If you have any questions in the meantime, please do not hesitate to write to me.

I should add that it is The EMBO Journal policy to allow only a single major round of revision and that it is therefore important to resolve these concerns at this stage. I believe the concerns of the referees are reasonable and addressable, but we are aware that many laboratories cannot function at full efficiency during the current COVID-19/SARS-CoV-2 pandemic, so please contact me if you have any questions, need further input on the referee comments or if you anticipate any problems in addressing any of their points. Please, follow the instructions below when preparing your manuscript for resubmission.

I would also like to point out that as a matter of policy, competing manuscripts published during this period will not be taken into consideration in our assessment of the novelty presented by your study ("scooping" protection). We have extended this 'scooping protection policy' beyond the usual 3 month revision timeline to cover the period required for a full revision to address the essential experimental issues. Please contact me if you see a paper with related content published elsewhere to discuss the appropriate course of action.

Again, please contact me at any time during revision if you need any help or have further questions.

Thank you very much again for the opportunity to consider your work for publication. I look forward to your revision.

Best regards,

William

William Teale, Ph.D.
Editor
The EMBO Journal

When submitting your revised manuscript, please carefully review the instructions below and include the following items:

- 1) a .docx formatted version of the manuscript text (including legends for main figures, EV figures and tables). Please make sure that the changes are highlighted to be clearly visible.
- 2) individual production quality figure files as .eps, .tif, .jpg (one file per figure).
- 3) a .docx formatted letter INCLUDING the reviewers' reports and your detailed point-by-point response to their comments. As part of the EMBO Press transparent editorial process, the point-by-point response is part of the Review Process File (RPF), which will be published alongside your paper.

4) a complete author checklist, which you can download from our author guidelines ([https://wol-prod-cdn.literatumonline.com/pb-assets/embo-site/Author Checklist%20-%20EMBO%20J-1561436015657.xlsx](https://wol-prod-cdn.literatumonline.com/pb-assets/embo-site/Author%20Checklist%20-%20EMBO%20J-1561436015657.xlsx)). Please insert information in the checklist that is also reflected in the manuscript. The completed author checklist will also be part of the RPF.

6) We require a 'Data Availability' section after the Materials and Methods. Before submitting your revision, primary datasets produced in this study need to be deposited in an appropriate public database, and the accession numbers and database listed under 'Data Availability'. Please remember to provide a reviewer password if the datasets are not yet public (see <https://www.embopress.org/page/journal/14602075/authorguide#datadeposition>). If no data deposition in external databases is needed for this paper, please then state in this section: This study includes no data deposited in external repositories. Note that the Data Availability Section is restricted to new primary data that are part of this study.

Note - All links should resolve to a page where the data can be accessed.

8) For data quantification: please specify the name of the statistical test used to generate error bars and P values, the number (n) of independent experiments (specify technical or biological replicates) underlying each data point and the test used to calculate p-values in each figure legend. The figure legends should contain a basic description of n, P and the test applied. Graphs must include a description of the bars and the error bars (s.d., s.e.m.).

9) We would also encourage you to include the source data for figure panels that show essential data. Numerical data can be provided as individual .xls or .csv files (including a tab describing the data). For 'blots' or microscopy, uncropped images should be submitted (using a zip archive or a single pdf per main figure if multiple images need to be supplied for one panel). Additional information on source data and instruction on how to label the files are available at .

10) We replaced Supplementary Information with Expanded View (EV) Figures and Tables that are collapsible/expandable online (see examples in <https://www.embopress.org/doi/10.15252/emboj.201695874>). A maximum of 5 EV Figures can be typeset. EV Figures should be cited as 'Figure EV1, Figure EV2' etc. in the text and their respective legends should be included in the main text after the legends of regular figures.

11) Our journal encourages inclusion of *data citations in the reference list* to directly cite datasets that were re-used and obtained from public databases. Data citations in the article text are distinct from normal bibliographical citations and should directly link to the database records from which the data can be accessed. In the main text, data citations are formatted as follows: "Data ref: Smith et al, 2001" or "Data ref: NCBI Sequence Read Archive PRJNA342805, 2017". In the Reference list, data citations must be labeled with "[DATASET]". A data reference must provide the database name, accession number/identifiers and a resolvable link to the landing page from which the data can be accessed at the end of the reference. Further instructions are available at .

Further instructions for preparing your revised manuscript:

We realize that it is difficult to revise to a specific deadline. In the interest of protecting the conceptual advance provided by the work, we recommend a revision within 3 months (19th May 2022). Please discuss the revision progress ahead of this time with the editor if you require more time to complete the revisions. Use the link below to submit your revision:

Referee #1:

This is a systematic analysis of the effect of phosphorylation of different domains of RBR on its function in regulating cell division, stem cell maintenance, and cell death in Arabidopsis root. RBR proteins play a central role coordinating cell division, cell differentiation and cell survival in development and in response to the environment. These distinct roles involve RBR interaction with multiple proteins. In this study, the authors dissected how phosphorylation of different domains and at some specific sites affect RBR1's functions and interactions with other proteins. The authors created phosphor-defective or phosphor-mimic mutations of phosphorylation sites in RBR's three domains (N, P, and C), and tested the abilities of the mutant proteins to rescue various phenotypes of the amiGO rbr mutants and to interact with proteins that interact with the wild type RBR1. The authors conclude that RBR's multi-functions rely on phosphorylation code and phosphorylation-independent mechanism in Arabidopsis thaliana. The results support the conclusion conceptually, as some distinct effects are observed for specific combinations of mutations. The manuscript reports important findings based on a large amount of work, and the transgenic plant materials generated in the study will be a useful resource for future studies. However, many results are complicated, making it unclear what the "code" is. Additional analyses seem necessary to provide clear associations between phospho-sites and function, i.e. to define the codes.

The main evidence for phospho-code comes from the distinct phenotypes of the [N-,P0, C-] and [N-,P-, C0] variants. [N-,P-, C0] over-complements the amiGO line and causes strong reduction in SCN and meristem division. This means that dephosphorylation of at least one residue in N and P domains is important for regulating cell division. Further analysis showed that T406 is the most important one in the N-term. [N-,P0, C-] complements the division phenotypes in SCN and meristem like the WT, but causes a dominant cell death phenotype. These results provide key evidence supporting that different combination of sites is important for cell division and survival.

I have the following main concerns:

1. While different combinations of mutations showed distinct effects, phospho-site mutations in a single domain have little effect. There is also a lack of opposite effects of the phospho-defective and mimin variants. Therefore, it is not clear what the "code" is. One possible problem is that their assays are rather non-specific (length of meristem, number of SCN cells...etc). It may be useful to include more qualitative cell-fate phenotypes, where RBR is known to play a role, like cortex-endodermis division, and stomata development. It would be interesting to examine if there is any stomata phenotype (number of stomates, clustering, four-lips, and stomata in guard cell).

2. The Y2H screen identified known and new RBR interactors, and many of them show different interactions with mutant RBR (Fig S5). Among these, POB1 is the only interactor with [N-,P0, C-], so it could in principle explain the unique behavior of this variant. But it is not brought up in the text, although there seems to be a link to programmed cell death:
<https://journals.plos.org/plosgenetics/article?id=10.1371/journal.pgen.1006540>
3. The variation of results for GRF5 is strange (Fig S5B). Did the other interactors (Fig S5A) show variation? I think at least 3 clones should be presented for all the interactors (Fig S5A) like in Fig S5B.
4. The T406 mutation. They need to justify this mutation better in the text. It is supposedly a CYCA3;4 target, but they basically do not explain this. They also don't really provide any context for this in the discussion.

Minor comments

p. 4 "phosphorylations in these domains have an additive effect". The effect is more than additive because each domain on its own has no effect, but combining both N and P strongly inhibits division.

Fig 6 legend. What is the cyan color? I assume RBR-cyan, but I don't find it in the legend.

Fig S2B: Diagram and label for last two rows are switched.

Fig S5: Panel labels don't match legend. The bottom panel has no label (D? legend describes B).

There are also some typos here and there, and some awkward English. Examples include:

Page 4: This late reduction in RBR levels bypasses is requirement in early developmental stages: "its" instead of "is"

Page 10: "variant by *tcx5/6* is weaker than in that observed in [N-,P-,NFC-]16": delete the first "in".

Referee #2:

The manuscript describes a study on the role of RBR phosphorylation at multiple sites of its three domains, N-, pocket and C-terminal by producing non-phosphorylatable and phospho-mimic mutations. The conclusion is that these phosphorylation events are largely cooperative, but there are some specificities towards three developmental readouts, stem cell divisions, meristematic formative divisions and cell death. The cell division arrest/differentiation effect with the multisite non-phosphorylatable RBR could be blocked by mutating the binding site on RBR to targets through LxCxE. They then used these mutant RBR forms to search for TF interactors that bind dependent or independent of phosphorylation or the presence of LxCxE motif. The manuscript represent a large body of work to study RBR phosphorylation in a multicellular developmental setting. The results largely consistent to what has been found in animal models, that the cell cycle regulatory function of RBR is tuned by the amount of phosphorylations along these three domains with different contributions to allosteric changes and thus interactions with E2Fs and some other targets. The mechanism for specificities is less clear in this study. Does it relies on monophosphorylations and interactions at specific sites within the three domains as it has been shown by Sanidas et al Mol Cell 2029? How these interactors would determine specificities? These questions would have been specifically interesting to pursue a bit further for the case of cell death, DNA damage response.

Specific comments

1. There are 16 potential CDK phosphorylation sites on the plant RBR, but to my knowledge only two has been experimentally confirmed, S911, T406. The human P-RBR antibody recognises S911 and would allow the characterisation how the other multisite mutations influence the phosphorylation of this site.
2. Not clear to me why they have chosen to study T406 single site phosphorylation as opposed to any of the other ones. Needs some explanation.
3. They have used own promoter constructs to generate transgenic lines, but the expression levels of the different constructs have not been tested. This might contribute to the varying strength of the phenotypes.
4. For stem cell proliferation the non-phosphorylatable N and P shows the highest level of repression while the C and P phosphomimetic have the highest derepression. This is an apparent contradiction and does not support their overall conclusion that the C domain has the greatest contribution.
5. Consistently, for the formative meristematic divisions the N and P domain has the largest effect, suggesting that the regulation is similar to the stem cells. I do not see an argument for P specificity here.
6. The root images for the phosphomimetic are missing (please include), but they have an interesting observation with the N0P0C+ have more cells yet no change in meristem size, meaning that the cell size must be affected, smaller. If this is indeed the case and specific for this line, it would be interesting to follow up or at least discuss.
7. It is not clear to me how they could conclude from the T406 mutation experiment the idea of combinatorial phosphorylation code? First, it appears to me that this construct is in WT background and thus results cannot be fully compared to the experiments with *amiGO* (not spelled out in the text), but the single mutant or full mutants do not appear to be that different. T406 might be an important site for all three readouts.
8. They show that cell death in *amiGO* can be blocked by all but not the N-P0C- and the N+P+C+ as well as the N406 (0) P0C- constructs. The 406 position might merit special attention in respect to cell death. They speculate that this implies two different

mechanisms for RBR-regulated cell death. It has been previously shown that CYCD3 o/e (that also leads to RBR hyperphosphorylation) is unable to trigger cell death. It would be interesting to know whether these two RBR mutants are compromised to form RBR foci in the nucleus upon genotoxic text, which might explain why they are compromised to protect against cell death.

9. The NF mutation constructs they introduced to the Col0 background (Fig6I). In terms of cell death this is uninformative, and would be better to see the effect in the amiGO background to reveal whether the pocket binding is important for this phenotype.

10. Surprisingly they could recover a fully non-phosphorylatable RBR expression line in the *tcx5/6* background and show that its expression reverts the overproliferation phenotypes in *tcx5/6*. They conclude that this supports the idea that RBR works through TCX and DREAM. I am not sure of this argument. Isn't it the case that in the lack of TCX the RBR construct should not have an effect if the two are in the same complex?

Referee #3:

RBR is a central regulator of cell division, differentiation and survival in plants. Like its animal counterpart pRb, RBR is regulated by phosphorylation at multiple sites and interact with multiple proteins. Previous studies with the animal pRb protein suggested a phosphorylation code, but experimental evidence for this is still lacking. Virtually nothing is known about the specific roles of phosphorylation sites in plants. In this study, Zamora-Zaragoza et al. addressed these questions systematically by creating domains with multiple mutations at phosphorylation sites and combining them using a GoldGate technology. In addition to lending support to the prevailing concept that an active, unphosphorylated RBR, is inactivated by phosphorylation, their studies revealed new findings that phosphorylation at different sites contributes to RBR function to a different extent and function differently in regulating cell division, differentiation and survival. Using the yeast-two hybrid method, they also showed that many proteins interact with RBR, although most of them are not affected by the phosphorylation status of RBR. These results shed light on the mechanisms by which RBR regulates a diverse range of biological processes. The experiments are generally well done and the manuscript is well written.

A caveat in this study is the protein-protein interaction studies, as yeast two-hybrid method is notoriously known for its high false positive rates. Another method, for example, BiFC should be used to validate some of the interacting partners, particularly those highlighted in the manuscript.

Another caveat is that the phosphor variants tested in this study may not be present in planta, because the in vivo phosphorylation status is not known. The authors pointed out this lack of knowledge, but they should be explicit about the limitation of their findings.

Other points

1. In some transgenic plants, the authors noticed more cells in the meristem but the meristem length was not significantly different than the wild type. Is the cells shorter? Otherwise, it is hard to understand. Please explain.
2. Is the *tcx 5/6* double mutant lethal? If so, how was the plants transformed?
3. In the 1st paragraph of the discussion, "the phenotypic strength of phospho-mimetic variants ranged between those observed for wild-type and amiGO (Figs. 2-4), which supports the prevailing conception that an active, unphosphorylated RBR, is inactivated by regulatory phosphorylations". I don't see the logic.
4. How to determine the boundary of the QC cells, CEI, and CSC counted in the meristem (many figures)? An outline in the figures may help. Similarly, it is unclear what cells are dead cells. They need to be marked in the image.
5. In the statistical analyses, why the number of seedling are so different for different mutant variations? Is 16 a good minimal number for some but not for others?

Response to Referees:

We like to take the opportunity to thank all three referees for their insightful and constructive comments. Below, we have addressed each of these comments in order of appearance.

Referee #1:

This is a systematic analysis of the effect of phosphorylation of different domains of RBR on its function in regulating cell division, stem cell maintenance, and cell death in Arabidopsis root. RBR proteins play a central role coordinating cell division, cell differentiation and cell survival in development and in response to the environment. These distinct roles involve RBR interaction with multiple proteins. In this study, the authors dissected how phosphorylation of different domains and at some specific sites affect RBR1's functions and interactions with other proteins. The authors created phosphor-defective or phosphor-mimic mutations of phosphorylation sites in RBR's three domains (N, P, and C), and tested the abilities of the mutant proteins to rescue various phenotypes of the amiGO rbr mutants and to interact with proteins that interact with the wild type RBR1. The authors conclude that RBR's multi-functions rely on phosphorylation code and phosphorylation-independent mechanism in Arabidopsis thaliana. The results support the conclusion conceptually, as some distinct effects are observed for specific combinations of mutations. The manuscript reports important findings based on a large amount of work, and the transgenic plant materials generated in the study will be a useful resource for future studies. However, many results are complicated, making it unclear what the "code" is. Additional analyses seem necessary to provide clear associations between phospho-sites and function, i.e. to define the codes.

The main evidence for phospho-code comes from the distinct phenotypes of the [N-,P0, C-] and [N-,P-, C0] variants. [N-,P-, C0] over-complements the amiGO line and causes strong reduction in SCN and meristem division. This means that dephosphorylation of at least one residue in N and P domains is important for regulating cell division. Further analysis showed that T406 is the most important one in the N-term. [N-,P0, C-] complements the division phenotypes in SCN and meristem like the WT, but causes a dominant cell death phenotype. These results provide key evidence supporting that different combination of sites is important for cell division and survival.

I have the following main concerns:

1. While different combinations of mutations showed distinct effects, phospho-site mutations in a single domain have little effect. There is also a lack of opposite effects of the phospho-defective and mimic variants.

Therefore, it is not clear what the "code" is. One possible problem is that their assays are rather non-specific (length of meristem, number of SCN cells...etc). It may be useful to include more qualitative cell-fate phenotypes, where RBR is known to play a role, like cortex-endodermis division, and stomata development. It would be interesting to examine if there is any stomata phenotype (number of stomates, clustering, four-lips, and stomata in guard cell).

We are thankful to the reviewer for the constructive feedback. We indeed previously suggested the notion of a 'phosphorylation code' from the observation that some RBR phospho-sites affect distinct phenotypes more than others, and that mutated phospho-sites in different domains have bigger phenotypic impact than mutations in the same domain, as shown with the T406- combinations. Therefore, RBR phosphorylation acts not only additive, but combinatorial as well. However, we fully acknowledge that 'cracking' such a code is out of our reach with the current evidence, as was also pointed out by referees 2 and 3. Thus, in the new version of our manuscript, we refrain from

referring to them as a 'code'. These changes are reflected in the title, abstract (line 33) and throughout the text (lines 86, 179, 206, 241, 396-398, 473, 494, 549).

We reason that opposite effects are expected if the phospho-sites in the Wt RBR are phosphorylated and dephosphorylated at similar rates. However, if a given phospho-site is more often in the phosphorylated state, the phospho-mimetic variant of it will show a Wt-like phenotype, whereas the phospho-defective variant will show a drastic phenotypic change. Moreover, except for the fully phospho-defective and fully phospho-mimetic variants, there are always a number of non-mutated phospho-sites influencing the phenotype, but whose phosphorylation status is unknown. Nevertheless, in some cases we do observe opposite effects: [N-,P-,CO]¹² and [N+,P+,CO]¹² significantly deviate the number of cells in the SCN from the Wt situation in opposite directions. On the other hand, the phospho-mimetic variants [N0,P+,C+]⁹ and [N+,P0,C+]¹¹ display SNC over-proliferation, and their phospho-defective counterparts, namely [N0,P-,C-]⁹ and [N-,P0,C-]¹¹, are likely to have an opposite effect (Fig EV3B,C), but because they are strongly pleiotropic and inviable at normal expression levels, we cannot add quantitative evidence. We discuss this point in lines 476 – 483.

We fully appreciate the suggestion of looking for qualitative evidence of the effect of phospho-sites, for example on reported stomatal phenotypes. Therefore, we set out to phenotype all the lines used in this study, but could not see any change from the wild-type phenotype. As shown in the Figure below, neither the phospho-variants nor the amiGO-RBR line display a mutant phenotype similar to those previously reported. We speculate that, since the best characterised role of RBR in the stomata development is to prevent the re-entry into cell division of the two guard cells (after being formed by the last symmetrical division of the Guard Mother cell), the phospho-defective variants (which in general are hyperactive forms of RBR), can only fulfill this role. Phenotypes like stoma in stomata or four lips are only expected in variants with low RBR activity like the Phospho-mimetic ones. Due to the lack of phenotypes in the amiGO-RBR genetic background, any possible effect of the phospho-mimetic variants is masked.

Figure. RBR phospho-variant plants display no observable phenotypes in stomata. Representative DIC micrographs of all lines used in this study show Wt-like stomata phenotypes. Numbers in panels are the number of aberrant phenotypes out of the total stomata scored. All aberrant phenotypes were one extra lip, or single lip stomata. We never observed the four lips nor the stoma in stomata phenotypes. The fraction of stomata clusters in relation to the total stomata scored was 0,02 for Col-0, 0,07 for amiGO, and all other lines ranged between 0,00 and 0,05. The last two panels show different stomata expressing free GFP, which was used as a marker for the expression of the amiGO construct.

We presume that the lack of phenotype in the amiGO line is influenced by the promoter used. In Matos et al 2014, where they described stomatal phenotypes associated to RBR downregulation, the amiRNA was expressed under the stomatal specific promoter pFAMA. We think that this setup allows a strong downregulation in the stomata without affecting other processes. In our case, the amiGO is driven by the 35S promoter, and very strong prolonged down-regulation of RBR would lead to pleiotropy and unviability. Therefore, it is possible that our 35S::amiGO line is not strong enough to uncover the stomatal phenotypes.

Extending the same line of reasoning, we postulate that stronger down-regulation of RBR would only exacerbate the phenotypic differences we observed in our RBR lines. Therefore, in spite of the negative results for our stomata experiment, and taking into account the cumulative evidence of the specific role of RBR in SCN cell number, meristem size, and cell death, we feel confident that the experiments presented in our manuscript are robust and reflect a clear role for the additive and combinatorial working of RBR phosphorylation. Future work could elucidate such a role in the stomatal lineage if different tools for obtaining a clean RBR null background are used.

2. The Y2H screen identified known and new RBR interactors, and many of them show different interactions with mutant RBR (Fig S5). Among these, POB1 is the only interactor with [N-,P0, C-], so it could in principle explain the unique behavior of this variant. But it is not brought up in the text, although there seems to be a link to programmed cell death:

<https://journals.plos.org/plosgenetics/article?id=10.1371/journal.pgen.1006540>

We thank the reviewer for pointing out this publication on the role of *N. benthamiana* POB1 in programmed cell death during the hypersensitive response. The specific binding to [N-,P0, C-]¹¹ and its role in PCD is now mentioned in our manuscript in lines 537 – 538.

3. The variation of results for GRF5 is strange (Fig S5B). Did the other interactors (Fig S5A) show variation? I think at least 3 clones should be presented for all the interactors (Fig S5A) like in Fig S5B.

Like the reviewer, we were puzzled about the variability of the interactions. We show three replicates of the GRF5 interactions precisely because it is the only interactor that showed such a variable result. For the rest of the interactions, the overall result is consistent with the representative clone.

4. The T406 mutation. They need to justify this mutation better in the text. It is supposedly a CYCA3;4 target, but they basically do not explain this. They also don't really provide any context for this in the discussion.

We agree with the reviewer that the decision to mutate T406 was not properly explained in the previous version of our manuscript. Now we explicitly mention that this decision was based on the conservation, knowledge about relevance of the corresponding phospho-site (T373) in the human homolog pRb (lines 247 - 251). We mention the preferential phosphorylation of T406 and S911 by CYCA3;4 together with other examples of specificity to discuss the possibility of a mechanism for combinatorial RBR phosphorylation orchestrated by diverse CYC-CDK complexes (lines 499 - 509).

Minor comments

p. 4 "phosphorylations in these domains have an additive effect". The effect is more than additive because each domain on its own has no effect, but combining both N and P strongly inhibits division.

We changed the wording to emphasize the combinatorial nature of this effect, leaving the additive effect as a possibility (Lines 158, 159).

Fig 6 legend. What is the cyan color? I assume RBR-cyan, but I don't find it in the legend.

This is now indicated in Figure legend 6 (line 1012).

Fig S2B: Diagram and label for last two rows are switched.

The diagram has now been corrected.

Fig S5: Panel labels don't match legend. The bottom panel has no label (D? legend describes B).

There are also some typos here and there, and some awkward English. Examples include:

Page 4: This late reduction in RBR levels bypasses is requirement in early developmental stages: "its" instead of "is"

Page 10: "variant by *tcx5/6* is weaker than in that observed in [N-,P-,NFC-]16": delete the first "in".

We appreciated much the detailed revision of the text and figures. All suggestions have been addressed in the new version of our manuscript.

Referee #2:

The manuscript describes a study on the role of RBR phosphorylation at multiple sites of its three domains, N-, pocket and C-terminal by producing non-phosphorylatable and phospho-mimic mutations. The conclusion is that these phosphorylation events are largely cooperative, but there are some specificities towards three developmental readouts, stem cell divisions, meristematic formative divisions and cell death. The cell division arrest/differentiation effect with the multisite non-phosphorylatable RBR could be blocked by mutating the binding site on RBR to targets through LxCxE. They then used these mutant RBR forms to search for TF interactors that bind dependent or independent of phosphorylation or the presence of LxCxE motif. The manuscript represent a large body of work to study RBR phosphorylation in a multicellular developmental setting. The results largely consistent to what has been found in animal models, that the cell cycle regulatory function of RBR is tuned by the amount of phosphorylations along these three domains with different contributions to allosteric changes and thus interactions with E2Fs and some other targets. The mechanism for specificities is less clear in this study. Does it relies on monophosphorylations and interactions at specific sites within the three domains as it has been shown by Sanidas et al Mol Cell 2029? How these interactors would determine specificities? These questions would have been specifically interesting to pursue a bit further for the case of cell death, DNA damage response.

We appreciate the positive and constructive comments of the reviewer on our manuscript. We acknowledge the lack of mechanistic insight connecting the phenotypes to the specific protein interactions that underly them. Besides specific protein interactions, some RBR interactors control specific processes due to their specific expression patterns, like SCARECROW and FAMA transcription factors (Cruz-Ramírez et al., 2012; Matos et al., 2014). We do like to point out that, with the Y2H screenings and the complementation of the *tcx5/6* double mutant by the fully phospho-defective RBR, our manuscript does provide a hint on the relevance of LXCXE motif-mediated interactions, specifically with members of the DREAM complex. As the phospho-defective RBR still has strong phenotypes in the *tcx5/6* background, or in combination with the NF mutation, it is clear that other non-LXCXE interactions are also important for phosphorylation-regulated RBR functions. We discuss this in lines 517-526. Yet we agree that these observations do not offer an explanation for the phenotypes observed only in specific phospho-variants, like the cell death response.

The cell death phenotype in relation to the DNA damage response is a rather complex process that cannot be easily explained. For example, in our related manuscript, we show that RBR and NAC044 have opposite independent effects on DNA damage-induced cell death, but the RBR-NAC044 protein interaction induces cell death (Zaragoza et al., 2024) ; but as shown in our Y2H screen, NAC044 binds to all phospho-variants, so it doesn't explain the specificity of the cell death-inducing phosphorylation variants. In response to the suggestion made by Referee 1 in point 2, we mention POB1 as another possible candidate for the cell death phenotype.

Lastly, we think that mono-phosphorylated isoforms of RBR might be important if the mechanism is conserved to what was reported by Sanidas et al. (2019). However, our results of mutating the highly conserved T406 site alone indicate this mono-phosphorylation has not much effect on it own. Moreover, Sanidas et al., (2019) tested 16 pRb isoforms (14 mono-phosphorylated, 1 fully phospho-defective, and 1 Wt pRb) and found a grand total of 438 pRb interactors, out of which 175 (40%) interact exclusively with Wt pRb. Because the emerging mono-phosphorylation code paradigm excludes intermediate phosphorylation states of pRb, they attributed these 175 to the hyper-phosphorylated pRb, making it the most active interactor of all isoforms. This observation is inconsistent with the fully phospho-mimetic RBR phenotypes resembling RBR depletion in the amiGO line. Thus, we believe that intermediate RBR phosphorylation isoforms cannot be discarded. We discuss this in lines 458 - 465.

Specific comments

1. There are 16 potential CDK phosphorylation sites on the plant RBR, but to my knowledge only two has been experimentally confirmed, S911, T406. The human P-RBR antibody recognises S911 and would allow the characterisation how the other multisite mutations influence the phosphorylation of this site.

We appreciate the suggestion of the reviewer to study the effect of distant phospho-sites on a particular site. Following this advice, we performed western blot analysis using the anti-Phospho-Rb (Ser807/Ser811), reported to cross-react with the plant RBR S911. Unfortunately, in our hands, the antibody recognized a band of aprox. 140 KDa in protein extracts of all phospho-variants (comprising the phospho-defective forms of all RBR sites), but also in the Col-0 and amiGO lines (see figure below), suggesting that the band does not correspond to the SCFP3A-tagged RBR variants nor the antibody recognized specific phosphorylation sites in the RBR protein. Since the molecular weight does not correspond to the endogenous RBR either, and these results were observed repeatedly in different technical and biological replicates, we refrain from drawing conclusions from this experiment.

Western blot analysis of Arabidopsis seedlings protein extracts carried out as described in the methods section. Upper panel, rabbit anti-Phospho-pRb Ser807/Ser811 antibody (Invitrogen, #PA5-17897) was used in a 1:500; middle panel, primary anti-GFP Polyclonal antibody (Thermo Fisher Scientific, A-6455) was used in a 1:1000 dilution; and the HRP-coupled goat anti-rabbit secondary antibody (Thermo Fisher Scientific, #31466) in a 1:5000 dilution for both panels. Lower panel, coomassie (Simply Blue Safe Stain, Thermo Fisher Scientific, #LC6065) of the transferred gel. The anti-Phospho-pRb Ser807/Ser811 antibody recognizes a band with similar molecular weight as the RBR-SCFP3A phospho-variants even in the Col-0 and amiGO samples which do not contain transgenic RBR-SCFP3A.

However, in the Table below we display the results of unpublished phospho-proteomics experiments where we validated that 12 out of the 16 phosphorylation sites are functional. For this experiment, an immunoprecipitation with anti-GFP was performed using root of transgenic Arabidopsis expressing pRBR::RBR-GFP, and phosphorylated peptides were enriched with Ti-IMAC.

Phosphosite	Number identified	Peptides	Number identified
T9	4	mEEVQPPVtPPIEPNGK	3
		mEEVQPPVtPPIEPNGKR	1
S375	10	IDALSsPAR	3
		KIDALSsPAR	3
		RKIDALSsPAR	4
		TFIsPLSPHK	1
S382	2	TFIsPLSPHK	1
S385	14	TFIsPLSPHK	1
		TFISPLSPHK	6
		TFISPLSPHKsPAAK	7
		TFISPLSPHKsPAAK	7
		TFISPLSPHKsPAAK	1
S389	11	TFISPLSPHKsPAAK	3
T406	11	IAAtPVSTAmTTAK	11
S665	6	HETcPGQNGGIRsPK	5
		HETcPGQNGGIRsPKR	1
S685	6	NSFTsPVK	3
		NSFTsPVKDR	3
S885	5	AVEANNKPEGQcPGsPK	5
S898	20	VSVFSPVPDmsPK	12
		VSVFSPVPDmsPCK	4
		VSVFSPVPDmsPCK	4
		KVSAVHNvYVsPLR	1
S911	16	KVSAVHNvYVsPLRGSK	3
		VSAVHNvYVsPLR	5
		VSAVHNvYVsPLRGSK	7
S942	5	SYAcVGESTHAYQsPSK	5

2. Not clear to me why they have chosen to study T406 single site phosphorylation as opposed to any of the other ones. Needs some explanation.

As discussed in response to point 4 of Referee 1, we have now added this explanation (lines 247 - 251).

3. They have used own promoter constructs to generate transgenic lines, but the expression levels of the different constructs have not been tested. This might contribute to the varying strength of the phenotypes.

We agree that the expression levels were only addressed by the intensity of the fluorescent protein tag. In the new version of our manuscript, we include a western blot analysis using the anti-GFP antibody that recognizes the transgenic SCFP3A-tagged RBR variants (Fig EV2A). This experiment shows a band at the expected molecular weight for the RBR-variants, with similar concentration variation as that observed by confocal microscopy. Moreover, this experiment makes it clearer that most but not all RBR phospho-variants accumulate to similar levels. We acknowledge that protein accumulation might have an effect on the phenotypes. However, the low accumulation of [N-,P0, C-]¹¹(Fig EV2A), [N0,P-, C-]⁹ (Fig EV3B,C) and [N-,P-, C-]¹⁶/tcx5/6 (Fig EV5E) contrasts with their high phenotypic strength, indicating that the phospho-sites substitutions have a greater influence on the specific phenotypes we measured than their protein accumulation (discussed in lines 137 – 139). We therefore can conclude that concentration differences have not led to erroneous interpretations of effects.

4. For stem cell proliferation the non-phosphorylatable N and P shows the highest level of repression while the C and P phosphomimetic have the highest derepression. This is an apparent contradiction and does not support their overall conclusion that the C domain has the greatest contribution.

We agree that this observation poses an apparent contradiction. As the reviewer points out, following that [N0,P+,C+]⁹ shows the highest de-repression, the corresponding phospho-defective [N0,P-,C-]⁹ should be the most repressive. However, despite the complication that this variant is not viable to properly quantify its effect, the T1 seedlings shown in Fig EV3B,C actually indicate that this variant is the most repressive one. This observation, although difficult to prove, could explain why we don't observe the strongest opposite effects in corresponding phospho-defective and phosphomimetic variants. As mentioned in the previous answer and also in response to Point 1 of Referee 1, [N-,P0,C-]¹¹ and [N0,P-,C-]⁹ have fertility defects that preclude their expression at comparable levels to other variants or their actual viability. To try to overcome this problem, we attempted to use an inducible system but without success.

5. Consistently, for the formative meristematic divisions the N and P domain has the largest effect, suggesting that the regulation is similar to the stem cells. I do not see an argument for P specificity here.

We agree that the phospho-defective [N-,P-,C0]¹² variant has the largest effect in restricting both SCN division and transit amplifying cells proliferation. But we have a separate argument that the P domain is specifically relevant for meristematic divisions because on its own, the [N0,P-,C0]⁵ variant has an effect on meristematic cell proliferation but not in the SCN; and no other domain on its own shows a similar specific effect. We have now explained this better in lines 190 – 193.

6. The root images for the phosphomimetic are missing (please include), but they have an interesting observation with the NOPOC+ have more cells yet no change in meristem size, meaning that the cell size must be affected, smaller. If this is indeed the case and specific for this line, it would be interesting to follow up or at least discuss.

We want to point out that the root images for all quantified lines are visible in panel A (Fig 3). As the reviewer indicates, a change in the number of cells without changing meristem size, is explained through cell size compensation. Such compensatory effects have been observed in cell cycle genes known to influence RBR phosphorylation. Following the reviewers advice, we discuss this point in the new version of our manuscript in lines 198 – 206.

7. It is not clear to me how they could conclude from the T406 mutation experiment the idea of combinatorial phosphorylation code? First, it appears to me that this construct is in WT background and thus results cannot be fully compared to the experiments with amigo (not spelled out in the text), but the single mutant or full mutants do not appear to be that different. T406 might be an important site for all three readouts.

This issue is in part addressed in response to point 1 of Referee 1. Regarding the T406 experiments, indeed, the genetic background was not properly specified. In our new manuscript version, besides the mentioning that “Unless otherwise noticed, amiGO-RBR was used as genetic background for transgenic plants” in the methods section (Lines 577, 578), we included a similar note in the figure legends 5 and 6 (lines 991 and 1013). As in previous figures, we compare these variants to Col-0 because we want to know whether each phospho-variant deviates from the wild type situation, as it was clear that the amiGO phenotypes are consistently different from all transgenic phospho-defective lines.

Our conclusion about the combinatorial nature of RBR regulation by phosphorylation derives from the observation that the cell death phenotype arises from the combination of the phospho-defective T406 module with the C- module only, but not with the P- module, nor by any of these modules on their own. Thus, it appears that none of the modules on their own are determining this phenotype, but the combination of them. Similarly, stem cell differentiation occurs in the combination of T06- with P-, similar to the combination of N- with P-, but none of them show this phenotype on their own (Fig2). Thus, we agree with the reviewer that T406 is important for all readouts, but this importance is at least partially influenced by the other sites with which it has been combined.

8. They show that cell death in amiGO can be blocked by all but not the N-POC- and the N+P+C+ as well as the N406 (0) POC-constructs. The 406 position might merit special attention in respect to cell death. They speculate that this implies two different mechanisms for RBR-regulated cell death. It has been previously shown that CYCD3 o/e (that also leads to RBR hyperphosphorylation) is unable to trigger cell death. It would be interesting to know whether these two RBR mutants are compromised to form RBR foci in the nucleus upon genotoxic stress, which might explain why they are compromised to protect against cell death.

We thank the reviewer for commenting on cell death phenotype. It is indeed very interesting that CYCD3 overexpression does not lead to cell death despite triggering other RBR-related phenotypes like SCN over-proliferation (Horvath et al., 2017). This observation resembles some of our phospho-mimetic variants such as [N0,P+,C+], [N+,P0,C+], and [N+,P+,C0], which all show SCN proliferation but not cell death (Figs 2 and 4). Thus, it is possible that CYCD3 o/e leads to phosphorylation of many but not all phospho-sites. We discuss this point in our new version of the manuscript lines 496 - 499.

We agree that studying foci formation upon genotoxic stress is very interesting, and we had actually thought about this previously. However, we decided not to go in this direction in our manuscript because our own research, partially published in (Zaragoza et al., 2024) showed us that foci formation does not imply protection. For instance, the N849F mutant rescued cell death in standard growing conditions but it was unable to form foci. Conversely, our own unpublished experiments with two RBR truncated variants showed us that they display cell death in standard growing conditions and are very susceptible to genotoxic stress despite forming foci (see figure below). In the figure below, we also show that the phospho- variants [N-,P0,C-]¹¹, and [N+,P+,C+]¹⁶, which show cell death in standard growing conditions, are able to form foci upon genotoxic stress. In our recently published manuscript, we emphasize that focus formation and cell death are related but belong to different stages of the DNA damage response, the foci being part of an early repair process, and the cell death constituting the later irreversible final cell fate decision (Zaragoza et al., 2024) . Thus, focus formation is not necessarily informative to our observations of the two phospho-variants showing cell death.

Figure. Foci formation is not indicative of protection against genotoxic stress and PCD. A) Cell death visualized by confocal imaging of longitudinal sections of PI-stained root tips 8 dpv on 0.5 GM medium; numbers indicate the amount of roots presenting dead cells in the truncated RBR variants RBR Δ N-YFP (where the whole N-domain is missing) and RBRpocket-YFP (only the AB-pocket domain without N- and C- regions). Data for amiGO, RBR-YFP and Col-0 is to be found in Fig 1 in our publication (Zaragoza et al., 2024). B) Root growth comparison of Col-0, amiGO, RBR-YFP, RBR Δ N-YFP and RBRpocket-YFP. 4 dpv seedlings were incubated on 10 μ M zeo for 16h and transferred again to 1/2GM for recovery. Root growth was scored for 6 dat. Data are presented as mean \pm SD ($n > 10$). *** $p < 0.001$, * $p < 0.05$. C, D) RBR Δ N-YFP and RBRpocket-YFP foci formation after a 16h 10 μ g/mL zeo treatment. Representative maximum-intensity projections of z-stack images from RBR-YFP, RBR Δ N-YFP and RBRpocket-YFP living roots nuclei (C) and quantification of foci (D) presented as mean \pm SD of foci number normalized by nuclei number. $n \geq 3$ roots, total nuclei per time >1300. E) Representative nuclei from living roots expressing RBR-sCFP and the phosphorylation variants [N-,P0,C-]¹ and [N+,P+,C+]¹⁶ showing foci after 16h incubation on 10 μ M zeo. $N > 5$ roots.

9. The NF mutation constructs they introduced to the Col0 background (Fig6I). In terms of cell death this is uninformative, and would be better to see the effect in the amiGO background to reveal whether the pocket binding is important for this phenotype.

We apologize for our lack of clarity on the genetic background. As mentioned in our response to point 7, we have now clarified that the NF mutant is also in the amiGO background. For more information about the effect of the NF mutation on PCD, please see manuscript (Zaragoza et al., 2024).

10. Surprisingly they could recover a fully non-phosphorylatable RBR expression line in the tcx5/6 background and show that its expression reverts the overproliferation phenotypes in tcx5/6. They conclude that this supports the idea that RBR works through TCX and DREAM. I am not sure of this

argument. Isn't it the case that in the lack of TCX the RBR construct should not have an effect if the two are in the same complex?

We agree with the reviewer that if the functions of RBR are mediated by the DREAM complex alone, the *tcx5/6* mutant should completely bypass the effect of the fully phospho-defective mutant. However, our Y2H experiment shows that phospho-defective RBR interacts with several proteins other than those found in the DREAM complex, and also with E2F proteins, which can be in the DREAM complex or not. Moreover, the partial rescue of phospho-defective variants by the NF mutation, which disrupts interaction with LXCXE proteins like TCX5/6, but not with E2F proteins (Cruz-Ramírez et al., 2013; Zaragoza et al., 2024), suggests that there are some LXCXE-independent functions. Therefore, we cautiously conclude that the dependency of RBR activity on the DREAM complex members TCX5/6 is partial (in lines 386 - 388).

Referee #3:

RBR is a central regulator of cell division, differentiation and survival in plants. Like its animal counterpart pRb, RBR is regulated by phosphorylation at multiple sites and interact with multiple proteins. Previous studies with the animal pRb protein suggested a phosphorylation code, but experimental evidence for this is still lacking. Virtually nothing is known about the specific roles of phosphorylation sites in plants. In this study, Zamora-Zaragoza et al. addressed these questions systematically by creating domains with multiple mutations at phosphorylation sites and combining them using a GoldGate technology. In addition to lending support to the prevailing concept that an active, unphosphorylated RBR, is inactivated by phosphorylation, their studies revealed new findings that phosphorylation at different sites contributes to RBR function to a different extent and function differently in regulating cell division, differentiation and survival. Using the yeast-two hybrid method, they also showed that many proteins interact with RBR, although most of them are not affected by the phosphorylation status of RBR. These results shed light on the mechanisms by which RBR regulates a diverse range of biological processes. The experiments are generally well done and the manuscript is well written.

A caveat in this study is the protein-protein interaction studies, as yeast two-hybrid method is notoriously known for its high false positive rates. Another method, for example, BiFC should be used to validate some of the interacting partners, particularly those highlighted in the manuscript.

We thank the reviewer for the constructive comments on our manuscript. It is indeed a common and good practice to confirm protein interactions by different methods. However, rather than providing mechanistic insights for the phenotypes, our Y2H experiment aimed to investigate whether all RBR protein interactions are regulated by phosphorylation, a long-assumed hypothesis but never proved directly for plant RBR and with a large number of interactors. Our results shows four exemptions (DREB2D, GBF4, NAC090, NAC044) to this hypothesis. From this experiment, RBR-TCX5/6 is the only interaction mechanistically relevant for our study, and it has been reported previously (Kobayashi et al., 2015; Lang et al., 2021; Ning et al., 2020).

We agree with the reviewer's concern of false positive interactions in Y2H. For instance, our ten different Y2H screenings fished out 28 interactors in total. However, we confirmed them using a high stringency that ruled out half of them. Even E2FC and XND1, well-known RBR partners, only displayed evident interaction with phospho-defective variants. Together with the fact that other known RBR interactors (FAMA, SCARECROW, ERF115, and E2FA) present in the library were not picked, we feel confident that our stringency conditions rule out false positives (but not false negatives). Moreover, we believe that this experiment constitutes a valuable resource for the scientific community in the

field, and to explore these interactions in detail has to be done in a case by case study, as we did for the NAC044 protein in our pre-print manuscript (Zaragoza et al., 2024), where the interaction is confirmed by Split-LUCIFERASE assay. In the new version of our manuscript, we include the Appendix Supplementary table 1 to explicitly mention the 14 out of 28 interactors that constitute possible false positives, and make this point clear in lines 346 - 349 and Appendix supplementary Figure 1 legend.

Another caveat is that the phosphor variants tested in this study may not be present in planta, because the *in vivo* phosphorylation status is not known. The authors pointed out this lack of knowledge, but they should be explicit about the limitation of their findings.

We agree with the referee that the exact phospho-sites mutation combinations we study are artificial and cannot reflect the *in vivo* phosphorylation/dephosphorylation dynamics. We now point this out explicitly and discuss how future research could approach it lines 470 - 476, and 485 - 491. Nonetheless, we hope that our approach is justified by the fact that most of the phospho-sites are functional *in vivo* and are not phosphorylated in equal degrees (see our answer to the Specific point 1 made by Referee 2, where the table shows variation in the abundance of phosphorylated sites).

Other points

1. In some transgenic plants, the authors noticed more cells in the meristem but the meristem length was not significantly different than the wild type. Is the cells shorter? Otherwise, it is hard to understand. Please explain.

Yes, these are compensatory effects as also explained above in response to point 6 of Referee 2, and in the new version of the manuscript in lines 198 - 206.

2. Is the *txc 5/6* double mutant lethal? If so, how was the plants transformed?

Although we noticed some fertility reduction, the *tcx5/6* double mutant is viable and it was transformed by floral dip. This is also reported in the original publication of the mutant line (Ning et al., 2020).

3. In the 1st paragraph of the discussion, "the phenotypic strength of phospho-mimetic variants ranged between those observed for wild-type and *amiGO* (Figs. 2-4), which supports the prevailing conception that an active, unphosphorylated RBR, is inactivated by regulatory phosphorylations". I don't see the logic.

We agree that the wording used in the previous version was unnecessarily confusing. In the new version, we re-phrase this sentence for more clarity as "*the phenotypic strength of phospho-mimetic variants ranged between those observed for Col-0 and *amiGO* (Fig 2-4), which supports the prevailing conception that phosphorylation reduces RBR activity*", and the changes are reflected in lines 400-402.

4. How to determine the boundary of the QC cells, CEI, and CSC counted in the meristem (many figures)? An outline in the figures may help. Similarly, it is unclear what cells are dead cells. They need to be marked in the image.

We did not count the number of dead cells, but scored the presence or absence of cell death per root. In the new version of our manuscript, we mention that "Red spots in the SCN area correspond to cell death, as PI selectively stains dead cells" in the figure legends for Fig 4, Fig EV2 and Fig EV5). Similarly, following the reviewers suggestion, we outlined the SCN region in Fig 2A, 5H, 6F.

5. In the statistical analyses, why the number of seedling are so different for different mutant variations? Is 16 a good minimal number for some but not for others?

We acknowledge the variation in the sample sizes. In general, we aimed for more than 12 seedlings per replicate. For a few genotypes we increased the sample size to have a more robust comparison with the control line. For others it was not always possible to reach the minimum due to the unavailability of viable seeds, as some genotypes have fertility or embryonic defects.

Dear Ben,

Thank you for submitting a revised version of your manuscript. We have now received re-review reports from all three referees, which I have included below. As you will see, you have addressed their concerns satisfactorily. Before I can finally accept the manuscript, there are some remaining editorial points which need to be addressed. In this regard would you please:

- include up to five keywords,
- include a 'disclosure and competing interests statement',
- rearrange the manuscript so that Fig. 4 is called out after Fig. 3; the legend for figure 4c is provided before legend of figure 4b, the legend for figure 5f, h is provided before legend of figure 5d-e, g, legend for figure 6f, h is provided before legend of figure 6d-e, g, this all needs to be rectified; there is a callout for Fig. 5L, but no such panel and callouts are missing for Fig. 6B and Supplementary Table 3 (which should be Appendix Table S3),
- include a title page in Appendix 1 and a table of contents with page numbers; use nomenclature Appendix Figure S1 / Appendix Table S1-S3 throughout the manuscript and Appendix PDF with the appropriate callouts,
- provide exact p values in the legends of figures 2b; 3b-c; 4b, d; 5d-e; 6d-e, g; EV 5f-h,
- in figure 5g, correct the mismatch between the annotated p values in the figure legend and the annotated p values in the figure file,
- define box plots in terms of minima, maxima, centre, bounds of box and whiskers, and percentile in the legends of figures 2b; 3b-c, 5d-e, i; 6d-e, g; EV 5f-g,
- define the nature of n is in the legends of figures EV 5f-g,
- define the scale bar for figures EV 4b-e and EV 4f-h,
- correct units of scale bar from μM to μm in the legends of figures 2a; 3a; 4a, c; 5c, f, h; 6a, c, f, h; EV 2b; EV 3a-k; EV 5d-e,
- rename movies to Movie EV1-EV2 with the corresponding callouts, zipping the legends with each movie file, and
- correct the email address of co-author Katinka Klap - k.klap@tue.nl

We include a synopsis of the paper (see <http://emboj.embopress.org/>). Please provide me with a general summary image, a two sentence statement and 3-5 bullet points that capture the key findings of the paper.

I am looking forward to receiving your revised manuscript.

EMBO Press is an editorially independent publishing platform for the development of EMBO scientific publications.

Best wishes,

William

William Teale, PhD
Editor
The EMBO Journal
w.teale@embojournal.org

We realize that it is difficult to revise to a specific deadline. In the interest of protecting the conceptual advance provided by the work, we recommend a revision within 3 months (7th Nov 2024). Please discuss the revision progress ahead of this time with the editor if you require more time to complete the revisions. Use the link below to submit your revision:

Referee #1:

The revision has adequately addressed my concerns and suggestions. this will be a good contribution to EMBO J.

I noticed "according" should be "according to" on line 92.

Referee #2:

Rb is a multiphosphorylation-dependent switch, regulation interactions with other proteins, but how the different phosphorylations contribute to Rb function in vivo is poorly understood in any system. This paper came out with an elegant system of combining phosphor-negative and phosphor-positive mutations in the three separate Rb domains and complementing the previously studies root phenotypes, stem cell maintenance, root meristem length and cell death. The results are well presented and sound.

The manuscript became much streamlined and have greatly improved compared to the previous submission many years ago.

Referee #3:

My concerns have been adequately addressed by the authors in their revised manuscript and response to reviewers' comments, and I think this study significantly advances our understanding of how the plant RB protein regulates cell divisions.

All editorial and formatting issues were resolved by the authors.

Dear Ben,

I am pleased to inform you that your manuscript has been accepted for publication in the EMBO Journal.

Congratulations! I'm really glad to see this work published.

Yours sincerely,

William

William Teale, PhD
Editor
The EMBO Journal
w.teale@embojournal.org
